

# Impact of snow deposition on major and trace element concentrations and fluxes in surface waters of Western Siberian Lowland

Vladimir P. Shevchenko[1], Oleg S. Pokrovsky[2], Sergey N. Vorobyev[3], Ivan V. Krickov[3], Rinat M. Manasypov[3,4], Nadezhda V. Politova[1], Sergey G. Kopysov[3], Olga M. Dara[1], Yves Auda[2], Liudmila S. Shirokova[2,4], Larisa G. Kolesnichenko[3], Valery A. Zemtsov[3], Sergey N. Kirpotin[3]

[1]Shirshov Institute of Oceanology RAS, 36 Nakhimovsky Pr., Moscow, Russia

[2] Geosciences Environment Toulouse, UMR 5563 CNRS, University of Toulouse, 14 Avenue Edouard Belin 31400, Toulouse, France

[3] BIO-GEO-CLIM Laboratory, Tomsk State University, 36 Lenina, Tomsk, Russia

[4] Institute of Ecological Problems of the North, 23 Nab Severnoi Dviny, RAS, Arkhangelsk, Russia

*Correspondence to*: Oleg S. Pokrovsky (oleg@get.obs-mip.fr)

**Abstract.** Towards a better understanding of chemical composition of snow and its impact on surface water hydrochemistry in poorly studied Western Siberia Lowland (WSL), dissolved (melted snow) and particulate (> 0.45 µm) fractions of snow were sampled in February 2014 across a 1700-km latitudinal gradient (c.a. 56.5 to 68°N) in essentially pristine regions. Concentration of dissolved Fe, Co, Cu, As, La, increased by a factor of 2 to 5 north of 63°N. The pH, Ca, Mg, Sr, Mo and U dissolved concentration in snow water increased with the increase in concentration of particulate fraction (PF), which was also correlated with the increase in calcite and dolomite proportion in the mineral fraction, suggesting an enrichment of meltwater by these elements during dissolution of carbonate minerals. The concentrations of Al, Fe, Pb, La and other insoluble elements in < 0.45 µm-filtered snow water decreased with the increase in PF. Principal Component Analyses revealed F1xF2 structure of major and trace element concentration in both dissolved and particulate fractions, with 2 factors not linked to the latitude. Sr, Mo, Sb, U, and, partially, Cu and Zn were most sensitive to the latitude of the sampling. The main sources of mineral components in PF are desert and semi-desert regions of central Asia.

Comparison of major and trace elements in dissolved fraction of snow with lakes and rivers of western Siberia across the full latitude profile revealed significant atmospheric input of a number of trace elements. The snow water concentration of DIC, Cl, $SO_4$, Mg, Ca, Cr, Co, Ni, Cu, Mo, Cd, Sb, Cs, W, Pb and U exceeded or were comparable with spring-time concentration in thermokarst lakes of the region. The spring-time river fluxes (May-June, representing the snow melt period) of DIC, Cl, $SO_4$, Na, Mg, Ca, Rb, Cs, metals (Cr, Co, Ni, Cu, Zn, Cd, Pb), metalloids (As, Sb), Mo and U in the discontinuous to continuous permafrost zone (64-68°N) can be fully explained by melting of accumulated snow. Therefore, the present study demonstrates significant and previously underestimated atmospheric input of many major and trace elements to their riverine fluxes during spring flood. The impact of snow deposition strongly increased northward, in discontinuous and continuous permafrost zones of frozen peat bogs, which is consistent with the decrease of the impact of rock lithology on river chemical composition in the permafrost zone of WSL, relative to the permafrost-free regions.





## 1 Introduction

The snow cover exhibits a number a properties making it unique natural archive and indicator of the ecosystem status
(Baltrėnaitė et al., 2014; Bokhorst et al., 2016; Boyarkina et al., 1993; Callaghan et al., 2011; Caritat et al., 1998, 2005;
Garbarino et al., 2002; Guéguen et al., 2016; Kashulina et al., 2014; Lisitzin, 2002; Niu et al., 2016; Ross and Granat,
1986; Singh et al., 2011; Siudek et al., 2015; Van de Velde et al., 1999; Vasilenko et al., 1985; Walker et al., 2003). The
snow washes out insoluble aerosols particles from the atmosphere as well as soluble compounds, including various
pollutants (Telmer et al., 2004; Barrie, 1986; Tranter et al., 1986, 1987). Unlike rain, the snow remains at the soil surface
and thus records all atmospheric input during the glacial period of the year. In boreal and subarctic regions, both dissolved
and particulate fraction of snow water reflect the chemistry of winter atmosphere, when the land is covered by snow and
the water surfaces are covered by ice. During winter, the input of mineral compounds from adjacent regions is minimal and
the main factor controlling chemical composition of snow is long-range, hundreds and thousands km, atmospheric transfer
(Franzén et al., 1994; Huang et al., 2015; Shevchenko, 2003, Shevchenko et al., 2000, 2010; Welch et al., 1991; Zdanowicz
et al., 1998, 2006; Zhang et al., 2015).
Numerous studies of particulate fraction of snow have been conducted in different regions including western Siberia
(Boyarkina et al., 1993; Ermolov et al., 2014; Golokhvast, 2014; Golokhvast et al., 2013; Kashulina et al., 2014; Makarov,
2014; Moskovchenko and Babushkin, 2012; Mullen et al., 1972; Salo et al., 2016; Shevchenko et al., 2002, 2010, 2016;
Talovskaya et al., 2014; Topchaya et al., 2012; Walker, 2005; Xu et al., 2016). The dissolved (< 0.45 µm or < 0.22 µm)
fraction of snow was traditionally studied in European subarctic (Caritat et al., 1998; Chekushin et al., 1998; Kashulina et
al., 2014; Reimann et al., 1999; Reinosdotter and Viklander, 2005) but the data on trace elements in snow water collected
in boreal, arctic and subarctic regions are limited. In contrast to numerous studies of trace element geochemistry of the
snow cover of high altitude zones of Asia and northern China (Dong et al., 2015; Kang et al., 2007; Lee et al., 2008; Wang
et al., 2015; Zhang et al., 2013), glaciers of Greenland (Barbante et al., 2003; Boutron et al., 2011; Candelone et al., 1996)
and Alaskan and Canadian High Arctic (Douglas and Sturm, 2004; Garbarino et al., 2002; Krachler et al., 2005; Snyder-
Conn et al., 1997), the trace element geochemistry of dissolved and particulate fraction of Siberian snow remains at the
beginning of exploration. This is especially true for large and geographically homogeneous territories of western Siberia,
presenting relatively similar level of snow deposition during winter seasons (i.e., from 100 mm of water in the south to
140–150 mm of water in the north) without any pronounced influence of large industrial centers, mountain regions and
marine aerosols over the territory close to 1.5 million km² (Resources, 1972, 1973; Boyarkina et al., 2013).
The originality of the present study consists in *i*) sampling of substantial (~1700 km) latitudinal transect in relatively
pristine zones comprising forest, forest tundra and tundra within the permafrost-free, discontinuous and continuous
permafrost regions; *ii*) assessment of both dissolved and particulate forms of major and trace elements in snow samples.
Given the scarcity of available measurements of snow chemical and particulate composition in Western Siberia, we aimed
at addressing the following specific issues: (1) characterizing the effect of the latitude on major and trace element
concentration in dissolved (< 0.45 µm) and particulate (> 0.45 µm) fractions of snow; (2) testing the link between
dissolved and particulate fractions of elements and the impact of particles mineralogy on snow chemical composition; (3)
comparing dissolved concentrations of major and trace elements in snow to those in lakes and rivers across the latitudinal
gradient of WSL and assessing the share of snow deposition on seasonal and annual export of dissolved elements by
western Siberian rivers. Via addressing quantitatively the abovementioned issues using unified  methodology in
unprecedentedly large geographical coverage (56 to 68°N) of orographically flat low populated terrain, we anticipate to
enhance our knowledge of the winter atmospheric deposition in western Siberia, in the absence of direct influence of





marine aerosols and large industrial centers and to foresee possible evolution of chemical composition of continental
waters subjected to change of atmospheric precipitation regime due to ongoing climate change.

**2. Study site, materials and methods**
**2.1. Geographic settings**

Western Siberia Lowland, located between the Ural mountains and the Yenisei River,  extends over 2000 km

from south to north and presents highly homogeneous, from physico-geographical point of view, taiga, forest-tundra
and tundra landscapes comprising bogs and mires in the permafrost-free zone and thermokarst lakes developed on flat
peat bogs (palsa) in the permafrost-bearing zone. Detailed physico-geographical description, hydrology, lithology and
soils can be found in earlier works (Botch et al., 1995; Smith et al., 2004; Frey and Smith, 2007; Beilman et al., 2009)
and in our recent geochemical studies (Manasypov et al., 2014; Stepanova et al., 2015; Pokrovsky et al., 2015, 2016).
Because of its exceptionally flat orographic context, extensive vegetation cover and relative remoteness from the Arctic
Coast (except the north of the Gyda and the Yamal peninsulas), the atmospheric precipitates in winter are likely to bear
the signature of remote desert and semi-desert regions of Central Asia. The anthropogenic impact is not strongly
pronounced because of $i$) low population density (average 6 people/km$^2$ but only 0.5–2 people/km$^2$ in the northern half
of WSL) and  $ii$) moderate local pollution from the gas burning in oil wells mostly in permafrost-free zone, south of the
Surgut town. The part of WSL north of 64°N contains essentially gas exploration facilities, which minimally impact the
environment. Taken together, latitudinal profile of the WSL presents a unique opportunity to study the chemistry of
atmospheric deposits within highly homogeneous physico-geographical context and relatively low local anthropogenic
impact.

**2.2. Snow sampling**

The snow of the WSL was sampled along the latitudinal transect S→N, from the vicinity of the Tomsk city (zone

of southern taiga) to the eastern coast of the Ob estuary (tundra zone) from 19.02.2014 to 5.03.2014 (**Fig. 1**). The possible
sources of snow deposition and the pathways of aerosols transport to the WSL were reconstructed by analyzing
meteorological maps and by calculating back trajectories of air transport to the observation points using NOAA's
HYSPLIT model (Draxler and Rolf, 2003). In order to assess a snapshot of snow deposition across 1700-km latitudinal
profile and collect the freshest snow that was subjected to minimal transformation, we chosen to sample only the upper
layer of the snow cover. This technique, in contrast to traditional sampling of full snow column allows adequate
representation of the upper fresh snow layer that had minimal transformation at the soil, and frequently used in remote
regions (Kang et al., 2007; Zhang et al., 2013). A previous study of isotope composition of collected snow proved its fresh
character, not subjected to any metamorphism (Vasil'chuk et al., 2016).

Upper 0–5 cm of snow was sampled in 39 locations (Fig. 1). All sampled points were located far than 500 m

from the winter road. Sampling was performed using metal-free technique, in protected environment, using pre-cleaned
plastic shovel and vinyl single-used gloves. Approximately 30 L of snow was collected into single-used polyethylene
bags. These polyethylene bags were thoroughly washed with 1 M HCl and abundant MilliQ water in the clean room
class A 10,000. In the laboratory, the snow was melted at ambient temperature, and filtered through pre-weighted

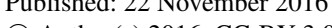



acetate cellulose filters (Millipore, 47 mm diameter) of 0.45 µm poresize. The storage of unfiltered snow water samples
was less than 1 h at 4°C.

**2.3. Particle analyses**

The sizes and morphology of particles on filters and elemental composition of individual particles were studied

using scanning electron microscope VEGA 3 SEM (Tescan) with microprobe attachment INCA Energy (Oxford
Instruments). The mineralogical composition of particulate fraction on selected filters was studied by X-ray powder
diffractometric method on the D8 ADVANCE (Bruker AXS) X-ray diffractometer equipped with the LYNXEYE linear
detector. The method of mineralogical composition characterization on filters is described elsewhere (Lisitzin et al.,
2015). The uncertainty of the relative proportion of mineral composition was 1–2% and the detection limit was 1%.

Freshly melted snow water was filtered through pre-weighted 0.45 µm acetate cellulose (Millipore) filters, placed

in the Petri dishes, dried at 60°C in an oven and fully digested using microwave acid attack which comprised 6.5 mL
concentrated $HNO_3$, 3.5 mL concentrated HCl and 0.5 mL concentrated HF. $HNO_3$ and HCl were bi-distilled in the clean
room and HF was commercial ultra-pure quality (Fluka). The filters were reacted 30 min in ultrasonic bath prior full
digestion using Mars 5 microwave digestion system (CEM, France). For this, 10 samples of filters, 1 certified 2711a
Montana II Soil standard and 1 blank filter sample were loaded into Teflon reactors subjected to treating at 150°C during
20 min. After completing the digestion, the content of reactors was transferred to 30 mL Savilex vials and evaporated at
70°C. The residue was dissolved in 10 mL of 10% $HNO_3$ and dilutes by 2% $HNO_3$ prior to the analyses. For the analysis of
snow particles on filters, the blanks (digestion of initial filters) were a factor of 10 to 100 lower than the filters containing
particles after 0.5-1.0 L of snow water filtration. The concentration of major and trace elements (TE) in filter digestion
products was measured using an ICP-MS Agilent 7500 ce with ~3 µg/L of indium and rhenium as internal standards.
Necessary corrections for oxide and hydroxide ion interferences were made for REEs and metals (Ariés et al., 2000). Based
on replicate analyses of in-house standards and certified materials, the uncertainty for TE measurement ranged from 5 % at
0.1–100 µg/L to 10 % at 0.001–0.01 µg/L. Analyses of low concentrations of Hf, Ge, Cs, Ga, and W (e.g., on the order of
0.001 µg/L, comparable with detection limits) was possible with a minimal estimated uncertainty of 20%. More details
about the whole procedure are available in Stepanova et al. (2014) and Pokrovsky et al. (2016).



**2.4. Melted snow analyses**

The pH and specific conductivity were measured on unfiltered snow water samples using Hanna portable

instruments. The dissolved (< 0.45 µm) fraction of snow water was obtained via filtration using a polycarbonate
Nalgene vacuum filter unit, and a PVC-made Mityvac hand vacuum pump. Blanks of MilliQ water were also placed in
polyethylene bags for the same time as melting snow (≤ 1 h at 4°C) and processed via filtration similar to snow samples.
The filtrates were divided into two parts; one was acidified with double distilled $HNO_3$ acid and stored in pre-cleaned
HDPE vials for ICP MS analysis, the second part was stored in HDPE bottles without acidification, for dissolved
organic and inorganic carbon analysis (DOC and DIC), respectively, and anion analysis by liquid chromatography.

The major anion concentrations ($Cl^-$, $SO_4^{2-}$) in the < 0.45 µm fraction were measured using ion chromatography

(HPLC, Dionex ICS 2000i), with an uncertainty of 2%, estimated from the replicate analyses of PERADE and RAIN
international certified materials. The DOC and DIC in this fraction were analyzed using a Carbon Total Analyzer
(Shimadzu TOC-VSCN) with an uncertainty of 5% and a detection limit of 0.1 and 0.05 mg/L, respectively.





Filtered snow water samples were analyzed with an Element XR ICP MS allowing for much better precision of
the analyses of highly diluted samples and avoiding many interferences of Agilent 7500 ce. The details of Element XR ICP
MS analysis in our laboratory are presented elsewhere (Pokrovsky et al., 2014). The international geostandards SLRS-5
(Riverine Water References Material for Trace Metals certified by the National Research Council of Canada) was used to
assess the validity and reproducibility of the analyses. For all major and most trace elements, the concentrations in the
blanks were below or comparable with analytical detection limits (≤ 0.1 ng/L for Cd, Ba, Y, Zr, REEs, Hf, Pb, Th, U; 1
ng/L for Ga, Ge, Rb, Sr, Sb; ~10 ng/L for Ti, V, Cr, Mn, Fe, Co, Ni, Cu, Zn, As). These values were at least 5 times lower
than the average concentration of trace elements in snow samples. Most TE presented in this work exhibited ≤ 15%-
agreement between the certified or recommended values and our measurements.  The TE  for which certified or
recommended data were available were considered only for the cases where we obtained good analytical reproducibility
(i.e., the relative standard deviation based on our standard measurements was ≤ 10%).


**2.5. River fluxes and snow storage**
The mass balance calculation of the degree of snow melt influence on element fluxes in WSL rivers were
performed taken into account *i)* the water stock in snow (in mm snow water accumulated during winter), fairly well
known for Western Siberia (Karnatzevitch and Khruschev, 2014; Resources, 1972, 1973; Zakharova et al., 2011) and *ii)*
the spring-time river runoff (in mm during May and June) calculated based on hydrological parameters. For water stock
calculation, we used the available mean multi-annual daily and monthly discharges  of WSL rivers across the latitudinal
profile (Resources, 1972 and 1972 and recently complied in the database R-AcricNET (www.r-arcticnet.sr.unh.edu).
Detailed description of WSL territory coverage by Russian Hydrological Survey (RHS) gauging stations and the
methods that were used to calculate the discharge during May-June are described elsewhere (Pokrovsky et al., 2015).
The most recent thorough hydrological measurements on small and medium size rivers of permafrost – affected part of
WSL (Novikov et al., 2009) were used together with RHS database were used to calculate the spring flood fluxes of
individual rivers and snow water stock for three latitudinal zones, 56-60°N, 60-64° and 64-68°N. Given that a
comparison between elementary snow stock and river runoff cannot be performed for individual river watersheds, since
no snow water chemical data are available with necessary spatial resolution, we compared the winter snow stock with
riverine spring flood fluxes of major and trace element for three latitudinal zones. For this, both spring flood fluxes of
individual rivers and snow water stock were averaged for each latitudinal zone.


**2.6. Statistical methods**
Statistical analysis of the average, median and geometric mean values and the link between element concentration
in suspended and dissolved fraction as well as comparison of different sampling sets (snow water and snow particles)
included ANOVA, H-criterion of the Kruskal–Wallis and Mann–Whitney U tests. These tests allowed evaluating the
difference between two sets of data separately for each TE following the approaches developed for lakes and rivers of
western Siberia (Manasypov et al., 2014, 2015; Pokrovsky et al., 2015, 2016). The multiple regressions was performed for
quantifying the relationship between dissolved and particulate concentration of TE and the latitudinal trends of
concentrations and enrichment factors. More thorough statistical treatment of both log-transformed and non-transformed
major and TE concentration in dissolved and particulate fraction of snow samples in each location included a normed PCA
analysis using the ADE-4 R package (Thioulouse et al., 1997) using the methods for scores and variables (De la Cruz and





Holmes, 2011). The data structure visualization was performed by presenting inertia ellipses regrouping the latitude and the
proximity to industrial centers (Chessel et al., 2004).

**3. Results**
**3.1. Soluble fraction of the snow water**
The latitude-averaged concentrations of dissolved and particulate fraction of snow samples are listed in **Table 1.**
A full data set of major and TE concentration in snow water is available at the location given in the "Data availability"
section. Examples of the effect of latitude on dissolved (< 0.45 μm) element concentrations are shown in **Fig. 2.** Fe and
Cu demonstrated a 2 to 5-fold increase in dissolved concentration north of 63°N (at $p < 0.05$). Zn and Pb did not exhibit
any systematic effect of latitude, and Sb, Cd and Ni demonstrated a single maximum at c.a. 63-65°N. As exhibited two
maxima, at 63.5 and 67.5°N with overall 2 to 3-fold decreasing trend northward. All other major and trace elements
demonstrated a lack of systematic variation of concentrations as a function of latitude (not shown).
The PCA treatment of soluble fraction suggested that at least two factors are interpretable. The PC1 x PC2
correlation circle revealed two groups of variables (**Fig. 3**). The first group is composed of Al, Fe, Cr, Zr, Pb, REEs
corresponding to lithogenic, poorly soluble trace elements. The second group is composed of DOC, K, Rb, Cs, Mn, Co,
Ba, Sb, Co, Mo, Mg, Si, Sr, Ca, pH. These highly mobile elements presumably reflect the marine aerosols and leaching
from soluble soil minerals such as carbonates as well as plant biomass. Similar factors determine chemical composition
of snow water regardless of the latitude of the sampling and no specific conditions or limiting factors depend on
geographical location. In order to identify the chemical elements most sensitive to the latitude, the linear regression
between the element concentration and the latitude was tested. The **Fig. S1** of the **Supplement** illustrates the evolution
of coefficient of determination, $R^2$, as a function of number of elements used in the regression. The effect of latitude can
be largely (96% of variability) explained by dissolved concentration of Cu, Zn, Sr, Mo, Sb and U.
The effects of particulate fraction on dissolved element composition in snow is illustrated in **Fig. 4** where the
value of pH (**4 A**), Sr (**4 B**), Al (**4 C**) and Pb (**4 D**) in dissolved fraction are plotted as a function of total particle
concentration in snow water. Ca, Mg, Sr, Mn, and Co increase their concentration in snow water by ca. an order of
magnitude with the increase in particle concentration by 2 orders of magnitude. The insoluble hydrolysates (Fe, Al, light
REEs, Zr, Cu and Pb) decrease their concentration (less than a factor 10) over two orders of magnitude increase in
particle concentration. Other elements in < 0.45 μm fraction exhibit the variations within an order of magnitude (DOC,
DIC, Na, Cl, $SO_4$, K, Si, Cr, V, Ni, Cu, Zn, As, Sb, Rb, Cd, Cs, Ba, heavy REE and U) or two orders of magnitude (Ti,
Ga, Mo, W) and do not demonstrate any significant (at $p < 0.05$) link with particle concentration.


**3.2. Possible impact of snow deposition on major and TE in lakes and rivers.**
3.2.1. Snow water in comparison to lake and river water
Chemical composition of dissolved snow water fraction can be compared with typical concentrations of major
and trace elements in thermokarst (thaw) lakes of western Siberia measured in 2013-2014. These lakes are shallow (0.5-
1.5 m depth) water bodies representing the largest reservoir of surface waters in western Siberia. They are widely
distributed north of 62°N where they may occupy 30 to 80% of the watershed area (Pokrovsky et al., 2014). The
average concentration of major and TE in thermokarst lakes of various size across significant latitudinal gradient taken
from Manasypov et al. (2014) can be compared with those in snow water collected in this study. Because the size of
thermokarst lakes of WSL ranges from few m² to several km², 4 representative ranges of lake diameters are used for this
comparison (0-10, 11-100, 101-500 and > 500 m). For this plot, only summer-period concentrations can be used as no




spring-time lake concentrations across the latitudinal gradient are available. Concentrations of low-soluble elements in
lakes are well above their concentrations in snow (**Fig. S2, A, B** for Al and Fe). At the same time, many trace elements
exhibited snow-water concentrations that were comparable or significantly higher ($p < 0.05$) than the concentrations in
lakes. Examples of Zn, Cu, Cd, Pb, Sb and Mo are given in **Fig. 5.** The excess of snow water concentrations over
summer lake concentration did not follow any particular latitudinal pattern.

Because the main source of water in shallow lakes of WSL in spring is melted snow (Manasypov et al., 2015),
we could compare the mean concentrations of snow water with spring-period lake water concentration for one particular
region of discontinuous permafrost zone (town of Nojabrsk, Khanymey site) for which high-resolution seasonal
observations on lakes of various size are available. For two classes of lake size ($< 0.5$ km² and $> 0.5$ km²), the following
three groups of elements could be distinguished. The concentrations of dissolved Na, Mn, Zn, As, Rb and Sr in snow
water are similar (within a factor of 2) to lake water concentrations. Concentrations of DIC, Cl, SO$_4$, Mg, Ca, Cr, Co,
Ni, Cu, Mo, Cd, Sb, Cs, W, Pb and U in snow are close or higher ($p < 0.05$) than those in lakes. And finally,
concentrations of DOC, Al, Si, K, Ti, V, Fe, Ga, Zr, Ba, and REEs in snow water are significantly lower that the lakes'
concentrations.

The concentrations of elements in snow water could be also compared with river water concentrations measured
during spring flood 2014 across the full latitudinal profile, since such data for rivers of different size are available for
the whole territory of WSL (Pokrovsky et al., 2015, 2016). Examples of elements whose concentrations in snow water
are higher or comparable with those in rivers during spring flood are illustrated in **Fig. S3**. Generally, the effect of snow
melt is mostly pronounced north of 64°N. During this period, when the rivers are essentially fed by melted snow, the
atmospheric deposition exhibited comparable or higher ($p < 0.05$) concentrations of SO$_4$, Cr, Co, Ni, Cu, Zn, Mo, Cd,
Sb, Cs, W and Pb than those in rivers. The concentrations of all other elements in WSL rivers cannot be explained by
solely snow water concentration.

3.2.2. Comparison of river fluxes in spring and snow water stock

Considering the mass balance calculation of snow melt influence on element fluxes in WSL rivers, the ratios of
river fluxes in May-June to snow stock are presented in the form of histograms (**Fig. S4 of Supplement**). These ratios
systematically decrease with the increase in the latitude which corresponds to the change of southern taiga to forest
tundra and tundra landscapes. In the southern, permafrost-free zone, Zn, Cd, W, Pb, Cs and Sb fluxes in rivers can be
provided essentially by snow melt. The riverine fluxes of DIC, Cl$^-$, SO$_4^{2-}$, Na, Mg, Ca, Sr, Rb, Cs, Zn, Cu, Cr, Ni, Cu,
Pb, As, Sb, Mo, W and U are strongly (i.e., $\geq 50\%$ at $p < 0.05$) affected by snow melt in the discontinuous and
continuous permafrost zones, north of 60-62°N.

According to the evolution of the ratio [river flux] / [snow stock] with the latitude, three groups of elements can
be distinguished: (*i*) elements that steadily decrease this ratio suggesting an increase in the impact of snowmelt
northward: DOC, SO$_4$, Al, Ti, V, Cr, Rb, Sr, Cd, Sb, Cs, La, Ce, W, Pb; (*ii*) elements for which this ratio decreases
abruptly to 62±2°N and then remains constant further northward: DIC, Na, Mg, Si,  K, Ca, Ni, Cu, As, Mo and U; (*iii*)
elements exhibiting non-systematic variation of the ratio with latitude but having strong ($> 50\%$) impact  of snowmelt
on river fluxes (Cl, Co, Zn, Ga) and (*iv*) elements having negligible ($< 10\ \%$) impact of snowmelt on river fluxes (Mn,
Fe, Zr and Ba). A summarizing histogram of elements whose riverine fluxes in spring are most affected by snow
deposition is given in **Fig. 6.** Overall, the impact of snow melt on river export fluxes in spring strongly increases
northward for DIC, Cl$^-$, SO$_4^{2-}$, Na, Mg, Ca, Cd, Pb, Sb, Cr, Cu, Ni, As, Mo, Rb, U.






### 3.3. Particles concentration and TE in particulate fraction of snow

Concentration of particulate fraction (PF) of snow and its elementary composition are at the location given in the "Data availability" section. The mineralogical composition of selected snow samples is given in **Table S1** of the Supplement. The dominant minerals are quartz (37%), albite (13%), K-feldspar (13%), phlogopite (10%), chrysotile (8%), illite (7%), and chlorite (5%). The concentration of dolomite and calcite ranges from 1 to 48 and 1 to 19%, respectively. Although mineral components dominated the composition of particulate fraction, the PF also contained organic fibers, diatom frustules, pollens and particles produced during fuel burning (fly ash and black carbon). Detailed morphological description of snow particles based on scanning electron microscope analysis is given elsewhere (Shevchenko et al., 2015). The concentration of particles in snow water ranged from 0.4 to 67 mg/L. The highest values are encountered in the vicinity of the Tomsk city (No SF 1) and around towns of Surgut (No SF 54, 14), Nojabrsk (SF 36, SF 38) and Gubkinsky (SF 33). Although the proportion of fly ash and black carbon in these samples is significant and higher than in the rest of samples as follows from SEM observation, the mineral particles (1-25 µm size) still dominate. Note that high content of fly ash and fuel burning spheres is not linked (p > 0.05) to high particulate and dissolved elements. The lowest concentrations of particles (< 5-10 mg/L) are recorded north of 65°N (the region of gas industry) and between 58 and 61°N (winter road along the Ob River with very low population density).

The PCA treatment of elementary composition of particulate fraction demonstrated the F1 x F2 structure (**Fig. S5** A of the Supplement). Here, two groups can be distinguished: highly mobile elements (Na, Ca, V, Ni, Mg, Mn) and low mobile elements (REE, Zr, Pb, Cd, Ga, P). In order to assess the degree of element fractionation in snow particles, Al-normalized TE enrichment factor (EF) with respect to the average upper part of continental earth crust (Rudnick and Gao, 2003) was calculated according to:

$$EF = \frac{[TE]/[Al]_{sample}}{[TE]/[Al]_{crust}}$$

The enrichment coefficient ranged from ~1–5 (Si, Ga, REEs, Fe) to > 100 (Mo, W, As, Sb, Ni, Cu, Pb, Mg, Ca, Na) as illustrated in **Fig. 7**. The highest enrichment (EF ≥ 1000) is observed of Sb, Zn and Cd. The variation of the enrichment factor as a function of latitude is shown for elements most enriched in particulate fraction in **Fig. S6 of Supplement.** For Mg, Ca, Sr, Ba, Fe, Mn, Co, Ni, K, Rb, Cs, V, Cr, As, Cd, W the EF exhibits a maximum around 63-64.5°N. This maximum coincides with the maximum of particulate fraction concentration (not shown).

The majority of chemical elements are present in particulate rather than dissolved form in snow meltwater samples. This is illustrated by a histogram of the ratios averaged over full latitudinal profile (**Fig. 8**). Although the variations of this ratio for different snow water samples across the WSL achieve ±1 order of magnitude, the average values shown in this figure illustrate the importance of particulate deposition of Al, Fe, Ga, REEs, Cr, V, Ti, Zr, Mo and W. For other elements, particulate and dissolved inputs in the form of snow are within the same order of magnitude. Some soluble elements such as Na, Cd, Ca, Sr, Ba, K, As and Zn exhibit the dominance of dissolved transport in snow.

### 4. Discussion

#### 4.1. Particulate versus dissolved transport of major and trace elements by snow

In accord with general knowledge of the aerosol chemistry of the Arctic (Barrie, 1986; Barrie and Barrie, 1990; Laing et al., 2014, 2015; Nguyen et al., 2013; Pacyna and Ottar, 1989; Shevchenko et al., 2003; Weinbruch et al., 2012), the principal component structure of chemical composition of dissolved snow fraction implies the combination of lithogenic source (dust and soil particles dissolution, providing low-mobile, insoluble elements such as Al, Fe, Cr, Zr, REEs) and marine aerosols (soluble forms, providing high concentration of mobile elements such as Ca, Mg, Na, Mo,





Ni). The latter may also originate from aeolian transport of carbonate-rich soils. The biogenic component may include
Mn, Zn, K, Rb, DOC, Si whereas the anthropogenic pollution originates from coal combustion (Sb, Co) and heating
systems, gas flaring at the gas oil production site as well as non-ferrous metal-smelter industry (Sb, Zn, Vinogradova et
al., 1993) and ground transportation (Pb, Cu, Zn, Cr, Ni, As, Rossini Oliva and Fernández Espinosa, 2007; Sutherland et
al., 2000).

The soluble highly mobile elements such as alkali and especially alkaline-earth elements, Sb, Mo, W and U

demonstrated an increase in their dissolved (< 0.45 μm) concentration with the increase in the total particulate fraction
(**Fig. 4 B**). We interpret this increase in concentration, also correlated with $pH_{snow\ water}$ increase (**Fig. 4 A**), as a result of
element leaching from soluble minerals such as calcite and dolomite. There is a positive ($R^2$ = 0.53, $p < 0.05$)
correlation between % of calcite in the particulate fraction of snow and Ca concentration in snow meltwater (not
shown). Therefore, we hypothesize that simultaneous mobilization of carbonate minerals and soluble elements from the
soil and rocks to the atmosphere occurs in southern, carbonate-rock bearing provinces where the winter aerosols are
generated. The generation of insoluble elements such as trivalent and tetravalent hydrolysates in dissolved fraction of
snow occurs independently of snow enrichment in solid particles. Indeed, the decrease, and not increase in insoluble
elements dissolved concentration with the increase in particle concentration (**Fig. 4 C, D**) suggests that these elements
are not desorbed or leached from mineral particles, either within the lieu of aerosol formation or during snow melting
and filtration in the laboratory.

The majority of measured elements are transported in particulate rather than dissolved fraction in the snow water

(**Fig. 8**). This is in general agreement with the results of other studies of snow deposition in Scandinavia and Kola
Peninsula (Reimann et al., 1996), north-eastern European Russia (Walker et al., 2003) and on drifting ice in the northern
Barents Sea (Gordeev and Lisitzin, 2005). An interesting particularity of dissolved fraction of snow in WSL is the
increase in soluble fraction of Fe, Cu and LREEs north of 63°N (**Fig. 2**). An increase in concentration of Fe, La, Ce,
northward has been also reported for rivers of the WSL sampled during this period of the year (Pokrovsky et al., 2016,
see also **Fig. S7 of** Supplement). We do not have a straightforward explanation for such a coincidence. Mobilization of
Fe-rich colloids, occurring in rivers of the northern part of WSL, is not expected to occur in the atmospheric aerosols,
since the DOC level in the latter is very low (0.9 ± 0.2 mg/L) and insufficient to stabilize dissolved Fe(III); besides, the
water surfaces of the north of WSL (thaw ponds and thermokarst lakes rich in dissolved Fe) are fully frozen in February
and thus cannot generate aerosols. Given that there is no enrichment in particular fraction of Fe and REE in snow
collected north of 63°N (not shown), leaching of Fe from snow particles to the soluble fraction of snow in the north of
WSL is also unlikely.

The enrichment of snow particulate fraction relative to the earth crust as shown by Al-normalized enrichment

coefficient (**Figs. 7, S6**) can be understood via taking into account particle concentrations in snow and their microscopic
observations (this study and Shevchenko et al., 2015, respectively). We suggest that enrichment of PF in clays supply
most trace elements. The atmospheric particles are capable exerting significant impact on soils and ground vegetation
(Kabata-Pendias and Pendias, 1984; Rasmussen, 1998; Steinnes and Friedland, 2006). In the case of WSL, the
elementary composition of snow particulate fraction was compared with three main reservoirs of elements within the
soil column, sampled over significant latitudinal profile, from 55°N to 68°N (Stepanova et al., 2015). These reservoirs
are averaged over full latitudinal range and include *i*) mineral fraction from the bottom of the peat column; *ii*) depth-
averaged peat column composition, and *iii*) *Sphagnum* mosses), collected in ombrotrophic bogs, which receive their
constituents essentially from the atmosphere (e.g., Santelman and Gorham, 1988). The elementary ratios of snow
particles to that in mineral soil, peat and moss of the WSL are illustrated in **Fig. 9 A, B, and C,** respectively. Given
significant uncertainties on the latitude-averaged values of element concentration in snow particles, mineral, peat and





moss of soil column, the deviation of the ratios from unity is significant if it exceeds a factor of 2 to 3. Compared to
mineral soil of WSL, the snow particles are strongly ($\geq 10\times$) enriched in Sb, Zn, Ni and Cd and in a lesser degree ($\geq 5\times$)
in Mg, Ca, Pb, Mo, and As (**Fig. 9 A**). Note that western Siberian soils, developed on sand and clay (silt) deposits
(Vasil'evskaya et al., 1986), are quite poor in Ca and Mg, especially in the permafrost-bearing zone north of 62°N. The
enrichment of snow particles relative to peat is observed for all elements, being particularly high ($> 50\times$) for Ni, Cr, Pb,
Cu, Zn, Mg, Na and Sb (**Fig. 9 B**). Only P, Ge and Cd, exhibiting high affinity to peat (Shotyk et al., 1990, 1992), are
not significantly ($p > 0.05$) higher in snow particles compared to the peat column. Finally, the mosses are most depleted
by all elements relative to snow PF with only biogenic elements (P, K, Rb, Mn and Cd) known to be concentrated in
bryophytes being non-significantly higher in snow particles relative to mosses (**Fig. 9 C**). The particularity of the
northern part of western Siberia lowland is that the active (seasonally unfrozen) soil layer is located within the organic
(moss+peat) rather than mineral horizon; the latter is represented by poorly reactive sands and clays (Baulin et al., 1967;
Baulin, 1985; Tyrtikov, 1973, 1979). As a result, the surface waters drain essentially organic part of the column which
is very poor in lithogenic elements (Pokrovsky et al., 2015, 2016). The supply of mineral particles from the snow
therefore may significantly enrich the rivers and lakes in dissolved alkaline earths, metal micronutrients, phosphorus
and other elements given high reactivity of incoming silicate and carbonate grains in acidic ($pH < 3\text{-}4$), organic-rich (10
$< DOC < 50$ mg/L) surface waters of Western Siberia. The degree to which such a supply can lead to overestimation of
the calculated chemical weathering export fluxes of cations in the permafrost zone is not possible to quantify.
Therefore, in view of the importance of atmospheric input of solid particles for mineral-poor, peat bogs of western
Siberia, seasonal, year-round measurements of particulate atmospheric deposition in this region are necessary.

**4.2. Effect of industrial centers and local pollution versus long-term transfer on the amount and chemical**
**composition of particulate fraction**
Regional background concentrations of dissolved metals in snow of Quebec, Canada are reported to be 1.1, 1.7,
and 1.6 $\mu g/L_{meltwater}$ for Cu, Pb, and Zn, respectively (Telmer et al., 2004). The values for Cu and Pb are comparable
with average snow water concentration across WSL (0.83 and 0.68, respectively) but the concentration of Zn in WSL
snow is significantly higher ($10.1\pm5.0$ µg/L, excluding 3 contaminated samples near the Tomsk city). Background
concentrations of dissolved Cu, Pb, and Zn in snow of Alaskan Arctic are much lower (0.08, 0.09 and 1.2, respectively,
Snyder-Conn et al., 1997). In snow from background areas of north-eastern European Russia concentrations of
dissolved Cu are near at the same level as in snow from WSL, concentrations of dissolved Pb and Zn are 2 times lower
(Walker et al., 2003). Concentrations of dissolved Cu and Zn in snow of NW Finland are few times lower than in snow
of WSL; concentrations of dissolved Pb are at the same level (Caritat et al., 1998).
Significant enrichment in Ni is known for the aerosols of the Arctic Ocean (Shevchenko et al., 2003) may be
linked both to Ni transport from Norilsk and Kola smelters but also with Ni fractionation at the sea surface (Duce et al.,
1976). Ni concentration in snow water of the northern part of WSL significantly exceeds that in the thermokarst lakes.
The winter snow stock of dissolved Ni is several times higher than the river export of this element during spring flood
in the permafrost-bearing zone of the WSL, north of 60°N, and Ni concentration in snow particles exceeds up to 2
orders of magnitude its concentration in moss and peat of the territory.
The winter-time deposition of dissolved ($< 0.45$ µm) metals on the surface of northern part of the WSL can be
calculated taking into account the mean multi-annual volume of accumulated snow during 8 winter months (in mm of
snow water) and the average concentration of elements in February snow collected north of 64°N. The monthly
depositions of selected metals ($\mu g \ m^{-2} \ month^{-1}$) on the north of the WSL in the form of snow are equal to 2.8, 12, 15,
210 and 0.9 for As, Ni, Pb, Zn, and Cd which is significantly higher than the values for winter deposition of insoluble





aerosols into the Russian Arctic (0.22, 0.74, 2.7, 1.3 and 0.056, respectively, Shevchenko et al., 2003). Only V exhibited
similar values of Arctic aerosol and snow deposition (0.71 and 0.96 µg m$^{-2}$ month, respectively).
The main source of mineral particles in the southern part of latitudinal profile (56–58°N) may be soils of steppe
and forest-steppe regions south of WSL, where the land is cultivated and the snow cover is relatively thin. The aeolian
transport of soil particles under these conditions may be efficient even in winter (Evseeva et al., 2003). The main source
of ash particles in southern part of the profile is the industry and transport of the city of Tomsk. There are numerous
studies of snow cover contamination by particles in the vicinity of Tomsk (Boyarkina et al., 1993; Yazikov et al., 2000;
Talovskaya et al., 2014). The concentration of particles in snow collected from 58°N to 61°N ranged between 0.85 and
5.72 mg/L which is comparable or slightly higher than the values reported for Arctic snow cover (Darby et al., 1974;
Mullen et al., 1972; Nürnberg et al., 1994; Shevchenko et al., 2002, 2010). It is important that in this zone of low PF
concentration, combustion spheres, fly ash and black carbon of few µm diameters were dominating. This can explain
relatively low concentration of all TE at low PF concentration, as carbon compounds likely contain very low proportion
of trace metals. The most important sources of fly ash and black carbon are known to be gas flaring in oil and gas
industry, land transport, heating plants, residential combustion, forest fires (mainly in summer), industrial plants (Bond
et al., 2004; Moskovchenko and Babushkin, 2012; Quinn et al., 2008; Stohl et al., 2013). Chemical pollution of
atmosphere during gas flaring associated with gas and oil industry is an important factor of anthropogenic impact on
atmospheric deposition in western Siberia (Raputa, 2013; Yashchenko et al., 2014). The black carbon produced during
gas burning is detected not only in western Siberia but in the Russian sector of the Arctic Ocean in high latitudes (Stohl
et al., 2013).
In the zone 62–64.5°N, where some impact of oil industry is possible, the concentration of insoluble particles in
snow were above 10 mg/L, achieving the value of 66.6 mg/L in sample SF36. Backward trajectories to this site using
Draxler and Rolf (2003) approach show that last few days before sampling air masses arrived from south-western
direction. Accordingly, the particulate fraction in these samples contained mostly mineral particles 1–25 µm size with
some fly ash (burning spheres), black carbon. It is possible that mineral particles are supplied here via long-range
transport from forest-steppe, steppe and semi-desert regions south and south-west from the study site. Indeed, during
winter snow coverage period, the dominant winds in this zone have S, SW and W directions (Moskovchenko and
Babushkin, 2012). The events of mineral dust transport over large distances are well known in the boreal zone (Lisitzin,
1978, 2011; Shevchenko et al., 2010).
Further north of studied latitudinal profile, from 65 to 68°N, the concentration of snow particles ranged from 0.8
to 9.2 mg/L. These values are within the background in the Arctic and subarctic (Darby et al., 1974; Mullen et al., 1972;
Nürnberg et al., 1994; Shevchenko et al., 2002). The particulate fraction was represented by mineral debris of 1 to 15
µm in size, with frequent but not significant presence of spherical ash particles, biogenic strains and porous carbon
particles. Because the main source of mineral particles is long-range transport from southern desert and steppe regions,
moving to the north decrease the influence of these provinces. Note here that rather "southern" point SF9 received the
air masses mostly from snow-covered regions as follows from backward trajectories calculations.

**4.3 Impact of dissolved fraction of atmospheric input on lakes and rivers TE composition.**
Quantitative comparison of element input to the land surface with winter snow and element export from the
watershed of WSL rivers provided the assessment of minimal atmospheric contribution to riverine fluxes. For this
comparison, we ignored: *i*) prior melt history of the snow cover, because the freshly fallen snow could be subjected to
minimal transformation; *ii*) the efficiency of snow sample to capture the total chemical load transferred from the



atmosphere into the snow pack, because only the snow water is postulated to contribute to the riverine flux; and *iii*) to
what extent the snowmelt interacts with the topsoil and vegetation litter through which the melt flows into the river.
A comparison of snow stock/river water fluxes demonstrates the increase in the influence of atmospheric
deposition northward (**Figs. 6 and S4 of Supplement**), whereas the chemical composition of the dissolved fraction of
snow, although subjected to significant variation, does not exhibit any systematic trend with the latitude as follows from
the PCA (section 3.1). The reason for this difference may be relatively low fluxes and concentrations in rivers of the
northern, permafrost-affected territory of the WSL compared to the southern, permafrost-free zone (Pokrovsky et al.,
2015, 2016). As a result, the impact of atmospheric deposition on the riverine transport is more pronounced in the
permafrost zone than in the permafrost-free zone. We expect this effect to be quite general for flat bog tundra areas of
northern Eurasia, including, in addition to northern part of western Siberia (~400,000 km²) studied in this work, the
Yamal and Gyda Peninsula (122,000 and 160,000 km², respectively), the North-Siberian Lowland (~700,000 km²), the
Kolyma Lowland (170,000 km²), and the Yana-Indigirka Lowland (180,000km²) with overall territory close to 1.7
million km². The impact of snow deposition on river elementary fluxes should be much lower in permafrost-bearing
mountainous terrain such as Central and Eastern Siberia, the Alaskan slopes, north of Scandinavian shield and Canadian
High Arctic. In those territories, two processes may decrease the contribution of snow deposition to river fluxes: 1) the
impact of local mineral dust for aerosols generation may be well pronounced and 2) the chemical weathering occurs
within the mineral seasonally unfrozen layer producing higher fluxes of inorganic components.
In contrast, in the lowlands of Northern Eurasia, the rivers drain essentially organic layer (peat bog) terrain, thus
mineral feeding of rivers is really low. As it is demonstrated in section 3.2.2 of this study, low chemical (cationic)
weathering in the north of the WSL during spring suggests that TDS_c and DIC fluxes in May-June in this and other
similar regions are essentially controlled by snowmelt, rather than by weathering. It follows that during the spring
period, the intensity of chemical weathering in these latitudes can be a factor of 2 (major cations) to 5 (TE) lower than
that deduced from riverine fluxes. However, given that the shares of spring flood period (May-June) in the annual
export fluxes are only 5 to 10% for major cations and 10 to 20% for TE (Pokrovsky et al., 2015, 2016), the overall
impact of atmospheric deposits on element export fluxes will be strongly pronounced (i.e., $\geq$ 50% of total measured
river flux value) only for elements which have the ratio of the spring-time river export to snow stock less than 0.2, i.e.,
$SO_4$, Cu, Mo, Cd, Sb, Cs, W and Pb.

**Conclusions**
The chemical composition of surface layer of snow cover was studied across a 1700-km latitudinal gradient in
western Siberia Lowland. The particulate fraction ranged from 0.4 to 66 mg/$L_{meltwater}$ and increased in the regions of
enhanced dust deposition from southern steppe and desert provenances, in the proximity of industrial centers and due to
fly ash production from gas burning of the oil exploration sites. The lowest concentrations of PF were measured in the
NW part of Tomskaya region and north of 66°N (< 10 mg/L, comparable with background values in the Arctic snow).
A PCA treatment of the elementary composition of snow water dissolved fraction demonstrated a F1 x F2 structure with
the first factor controlling the insoluble, low-mobile lithogenic elements and the 2$^{nd}$ factor acting on alkaline-earth
metals, biogenic elements and anions. None of the factors was linked to the latitude.
There was an increase in concentration of soluble elements in snow water (Ca, Mg, Sr, Mn, Co) and pH with the
increase in mineral (calcite, dolomite) fraction. The concentration of insoluble elements (trivalent and tetravalent
hydrolysates, Cr, Pb, Cd, Cu) in snow water did not change or decreased in response to the increase in PF. The
elementary composition of PF demonstrated its significant enrichment in most elements relative to mineral soil horizon,





peat and moss composition averaged across full latitudinal profile (~1700 km) of WSL. As such, solid atmospheric
aerosols may be important factor of insoluble element delivery to the soil surface.
The impact of the snowmelt on chemical composition of western Siberian thermokarst lakes may be very high.
This will be further accentuated by reported increase in the proportion of meltwater that does not reach the main rivers
but stored by the wetlands (i.e., from 20-30% in early 1990s to 50-60% in the mid-2000s, Zakharova et al., 2011). The
concentrations of Na, Mn, Zn, As, Rb and Sr in winter aerosols are similar (within a factor of 2) to lake water
concentrations during spring period. Concentrations of DIC, Cl, $SO_4$, Mg, Ca, Cr, Co, Ni, Cu, Mo, Cd, Sb, Cs, W, Pb
and U in filtered snow water are close or higher than those in lakes. In the southern, permafrost-free zone, only Zn, Cd,
W, Pb, Cs and Sb fluxes in rivers during May-June period can be supplied essentially by dissolved fraction of the snow
melt. However, the impact of snow melt on river export fluxes in spring strongly increases northward for DIC, Cl, $SO_4$,
Na, Mg, Ca, Cd, Pb, Sb, Cr, Cu, Ni, As, Mo, Rb, U. In the permafrost zone, $\geq$ 50% of riverine fluxes of these elements
during spring flood can be provided by the snowmelt. The reason for such high sensitivity of WSL surface reservoirs to
atmospheric deposition is feeding of surface waters by essentially organic (moss, peat) soil profiles where very thin
layer of sand and clay horizons are subjected to seasonal thaw. With further increase of winter precipitation in western
Siberia (i.e., Bulygina et al., 2009), the impact of snowmelt on element transport to the Arctic Ocean by rivers  may
increase thus enriching the surface waters in many elements such as Cd, Pb, Sb, Cr, Cu, Ni, As, Mo, Rb, U. The snow
deposition of mineral particles on the moss cover developed over the frozen peat in the north of WSL will be mostly
pronounced for  Sb, Zn, Ni and Cd and in a lesser degree for Mg, Ca, Pb, Mo, and As, since these elements are
impoverished in mineral horizons of the WSL. However, quantifying the degree of these changes require a year-round
monitoring of liquid and solid atmospheric deposits across the WSL territory. We foresee a possibility to apply the mass
balance calculations of atmospheric input to the land surface of other Siberian lowlands of peat bog and thermokarst
lake zones, with an overall territory close to 1.7 million km².

**Data availability**
Full data set of major and trace element concentration in snow water (< 0.45 μm) and snow particles sampled
across   the   latitudinal   profile   of   Western   Siberia   Lowland   is   available   at   the   Research   Gate,
https://www.researchgate.net/publication/309666956; DOI: 10.13140/RG.2.2.12156.54408.

**Acknowledgements:**
This work was supported from the BIO-GEO-CLIM grant No 14.B25.31.0001 of Russian Ministry of Science and
Education. RM and LS acknowledge support from RSCF (RNF) grant No 15-17-10009. Supports from GDRI CAR-WET-
SIB, JPI project "SIWA" and Program 32 of Fundamental Research of Presidium of Russian Academy of Sciences are also
acknowledged. We would like to thank Academician A.P. Lisitzin for valuable recommendations, J. Prunier, M. Henry, A.
Lanzanova for help in analytical work. The authors acknowledge the NOAA Air Resources Laboratory (ARL) for the
provision of the HYSPLIT transport model and READY website (http://www.arl.noaa.gov/ready.html).



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

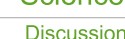
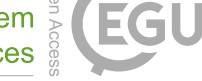


**Table 1.** Minimal, maximal, median and geometric mean concentration of dissolved ($\mu g\ L^{-1}_{snow\ water}$), n=35, and particulate
($\mu g\ g^{-1}_{particles}$), n=34 snow components. The data for upper continental crust (UPC) are from (Rudnick and Gao, 2003). N.A.
stands for non analyzed.

| Element | Dissolved | | | | Particulate | | | | UPC |
|---|---|---|---|---|---|---|---|---|---|
| | Min | Max | Median | Geometric mean | Min | Max | Median | Geometric mean | |
| pH | 4.38 | 8.73 | 5.11 | 5.44 | N.A. | N.A. | N.A. | N.A. | N.A. |
| SC, $\mu S\ cm^{-1}$ | 9 | 35 | 15.5 | 16.3 | N.A. | N.A. | N.A. | N.A. | N.A. |
| DIC, mg/L | 0.26 | 2.12 | 0.37 | 0.47 | N.A. | N.A. | N.A. | N.A. | N.A. |
| DOC, mg/L | 0.46 | 1.87 | 0.84 | 0.85 | N.A. | N.A. | N.A. | N.A. | N.A. |
| Cl, mg/L | 0.07 | 2.94 | 0.51 | 0.48 | N.A. | N.A. | N.A. | N.A. | N.A. |
| $SO_4$, mg/L | 0.41 | 2.01 | 0.71 | 0.72 | N.A. | N.A. | N.A. | N.A. | N.A. |
| Li | N.A. | N.A. | N.A. | N.A. | 2.6 | 32.2 | 10.8 | 10.7 | 24 |
| Be | N.A. | N.A. | N.A. | N.A. | 0.12 | 2.11 | 0.59 | 0.59 | 2.1 |
| Na | 47 | 1982 | 295 | 303 | 1452 | 39156 | 6717 | 7314 | 24200 |
| Mg | 19 | 862 | 114 | 114 | 3492 | 156712 | 19089 | 21411 | 14900 |
| Al | 1.6 | 35.2 | 15.5 | 12.3 | 6444 | 138267 | 31079 | 31565 | 81500 |
| P | N.A. | N.A. | N.A. | N.A. | 70 | 1928 | 481 | 503 | 660 |
| Si | 3.5 | 180 | 64.6 | 33.2 | N.A. | N.A. | N.A. | N.A. | N.A. |
| K | 39.2 | 120 | 55.5 | 63.0 | 1682 | 38395 | 5895 | 6023 | 23200 |
| Ca | 57 | 2266 | 267 | 296 | 3944 | 159272 | 17331 | 17775 | 25600 |
| Ti | 0.001 | 0.338 | 0.032 | 0.018 | 194 | 5762 | 674 | 689 | 3800 |
| V | 0.007 | 0.221 | 0.051 | 0.049 | 23.8 | 322 | 67.4 | 69.7 | 97 |
| Cr | 0.027 | 0.340 | 0.111 | 0.117 | 43.8 | 841 | 138 | 156 | 92 |
| Mn | 0.62 | 9.54 | 3.06 | 2.99 | 180 | 1242 | 400 | 404 | 780 |
| Fe | 1.8 | 62.2 | 14.6 | 12.0 | 7206 | 41255 | 15873 | 16488 | 39100 |
| Co | 0.006 | 0.418 | 0.097 | 0.094 | 5.9 | 60.7 | 19.4 | 18.6 | 17.3 |
| Ni | 0.04 | 5.66 | 0.36 | 0.36 | 28.1 | 1067 | 149 | 145 | 47 |
| Cu | 0.16 | 2.51 | 0.57 | 0.63 | 13.1 | 273 | 63.4 | 75.1 | 28 |
| Zn | 1.7 | 31.0 | 8.3 | 8.3 | 70.7 | 3832 | 202 | 255 | 67 |
| Ga | 0.0001 | 0.0185 | 0.0023 | 0.0014 | 1.8 | 26.7 | 8.32 | 7.73 | 17.5 |
| Ge | N.A. | N.A. | N.A. | N.A. | 0.36 | 3.18 | 0.88 | 0.91 | 1.4 |
| As | 0.02 | 0.46 | 0.19 | 0.15 | 3.8 | 67.2 | 16.1 | 16.0 | 4.8 |
| Rb | 0.033 | 0.262 | 0.066 | 0.075 | 6.1 | 124 | 24.1 | 23.4 | 84 |
| Sr | 0.26 | 10.2 | 1.04 | 1.23 | 26.2 | 580 | 117 | 115 | 320 |
| Y | N.A. | N.A. | N.A. | N.A. | 1.52 | 40.2 | 7.1 | 7.5 | 21 |
| Zr | 0.0001 | 0.0403 | 0.0024 | 0.0015 | 7.7 | 383 | 36.8 | 38.5 | 193 |
| Nb | N.A. | N.A. | N.A. | N.A. | 0.80 | 25.5 | 3.54 | 3.72 | 12 |
| Mo | 0.0001 | 0.059 | 0.010 | 0.005 | 0.55 | 10.4 | 2.12 | 2.24 | 1.1 |
| Cd | 0.015 | 0.180 | 0.047 | 0.046 | 0.11 | 3.37 | 0.71 | 0.71 | 0.09 |





| | | | | | | | | | |
|---|---|---|---|---|---|---|---|---|---|
| **Sn** | N.D. | N.D. | N.D. | N.D. | 1.13 | 29.3 | 7.76 | 7.36 | 2.1 |
| **Sb** | 0.009 | 0.132 | 0.036 | 0.038 | 1.67 | 27.2 | 5.96 | 6.15 | 0.4 |
| **Cs** | 0.0015 | 0.0105 | 0.0034 | 0.0036 | 0.32 | 4.78 | 1.35 | 1.24 | 4.9 |
| **Ba** | 0.74 | 13.6 | 3.35 | 3.32 | 88 | 1664 | 374 | 391 | 628 |
| **La** | 0.001 | 0.049 | 0.012 | 0.011 | 2.0 | 60.2 | 10.3 | 10.8 | 31 |
| **Ce** | 0.003 | 0.095 | 0.022 | 0.019 | 4.05 | 128 | 19.0 | 20.6 | 63 |
| **Pr** | 0.0001 | 0.0084 | 0.0022 | 0.0019 | 0.50 | 15.5 | 2.30 | 2.35 | 7.1 |
| **Nd** | 0.0013 | 0.0275 | 0.0085 | 0.0067 | 1.86 | 58.6 | 8.32 | 8.70 | 27 |
| **Sm** | 0.0001 | 0.0072 | 0.0020 | 0.0016 | 0.39 | 11.8 | 1.78 | 1.79 | 4.7 |
| **Eu** | 0.00010 | 0.00253 | 0.00096 | 0.00083 | 0.11 | 2.56 | 0.45 | 0.47 | 1.0 |
| **Gd** | 0.0004 | 0.0082 | 0.0022 | 0.0022 | 0.40 | 10.3 | 1.71 | 1.77 | 4.0 |
| **Dy** | 0.00002 | 0.0041 | 0.0016 | 0.0008 | 0.32 | 7.83 | 1.35 | 1.42 | 3.9 |
| **Ho** | 0.00006 | 0.00123 | 0.00061 | 0.00054 | 0.06 | 1.51 | 0.26 | 0.27 | 0.83 |
| **Er** | 0.0002 | 0.0029 | 0.0010 | 0.0010 | 0.18 | 4.71 | 0.77 | 0.80 | 2.3 |
| **Tm** | 0.00002 | 0.00088 | 0.00011 | 0.00009 | 0.03 | 0.72 | 0.11 | 0.11 | 0.3 |
| **Yb** | 0.00000 | 0.00289 | 0.00089 | 0.00049 | 0.16 | 4.91 | 0.73 | 0.73 | 1.96 |
| **Lu** | N.A. | N.A. | N.A. | N.A. | 0.024 | 0.76 | 0.11 | 0.11 | 0.31 |
| **Hf** | N.A. | N.A. | N.A. | N.A. | 0.25 | 13.2 | 1.10 | 1.18 | 5.3 |
| **Ta** | N.A. | N.A. | N.A. | N.A. | 0.18 | 4.35 | 0.62 | 0.62 | 0.9 |
| **W** | 0.002 | 0.108 | 0.020 | 0.017 | 2.0 | 102 | 35.9 | 28.8 | 1.9 |
| **Tl** | N.A. | N.A. | N.A. | N.A. | 0.04 | 0.73 | 0.23 | 0.23 | 0.90 |
| **Pb** | 0.02 | 3.67 | 0.51 | 0.38 | 13.2 | 703 | 71.9 | 67.9 | 17 |
| **Th** | N.A. | N.A. | N.A. | N.A. | 0.43 | 17.1 | 2.22 | 2.33 | 10.5 |
| **U** | 0.0007 | 0.0063 | 0.0031 | 0.0028 | 0.19 | 4.69 | 0.92 | 0.93 | 2.7 |







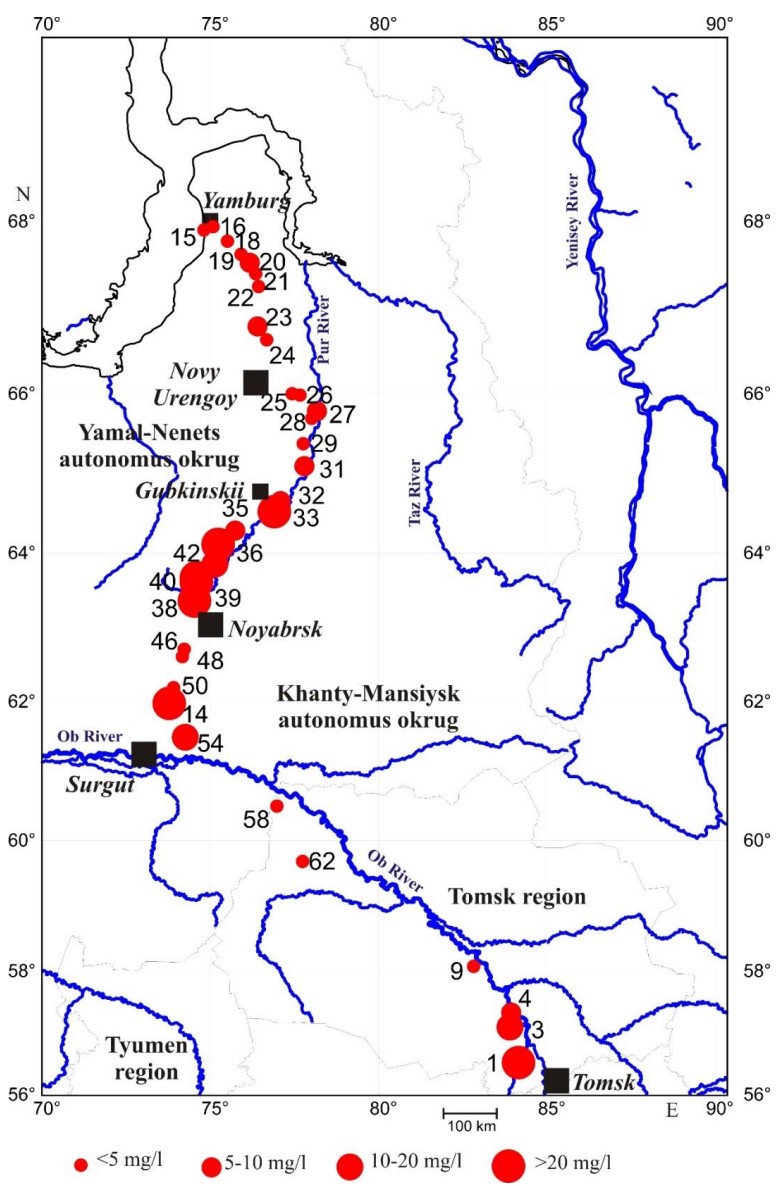



**Figure 1.** Map of the study site. The size of the sampling points reflects the concentration of particulate fraction
(mg/L$_{snow\ water}$)



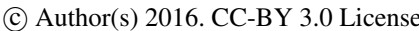





**Fig. 2.** Examples of dissolved (< 0.45 µm) metal concentrations in snow water as a function of latitude.





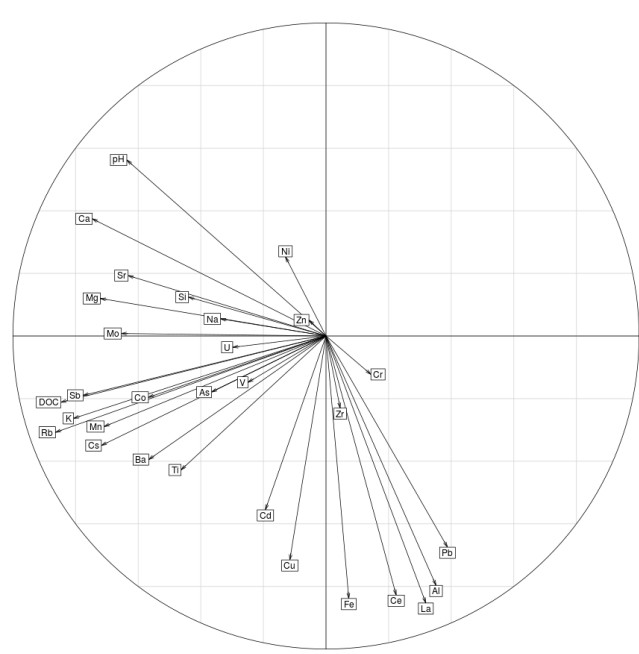

dissolved part
**Figure 3.** PCA Factorial map F1xF2 of variables (elements) for the dissolved fraction.













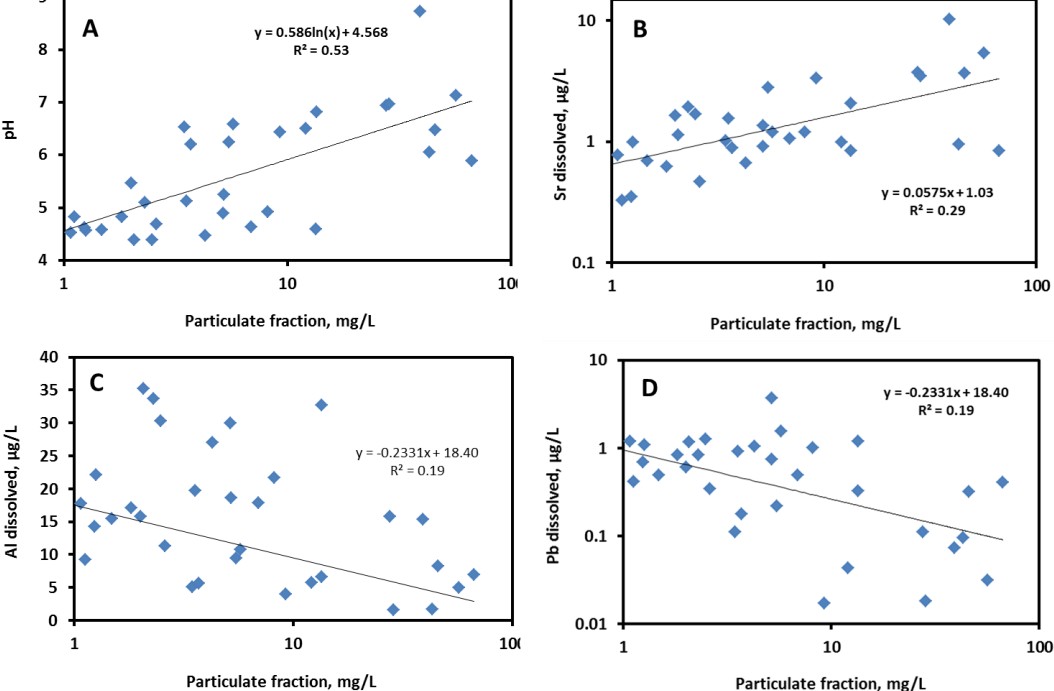




**Figure 4.** pH value (A) and Sr (B), Al (C) and Pb (D) concentration in dissolved fraction of snow as a function of concentration of particles. Note log X scale for Sr and Pb.

















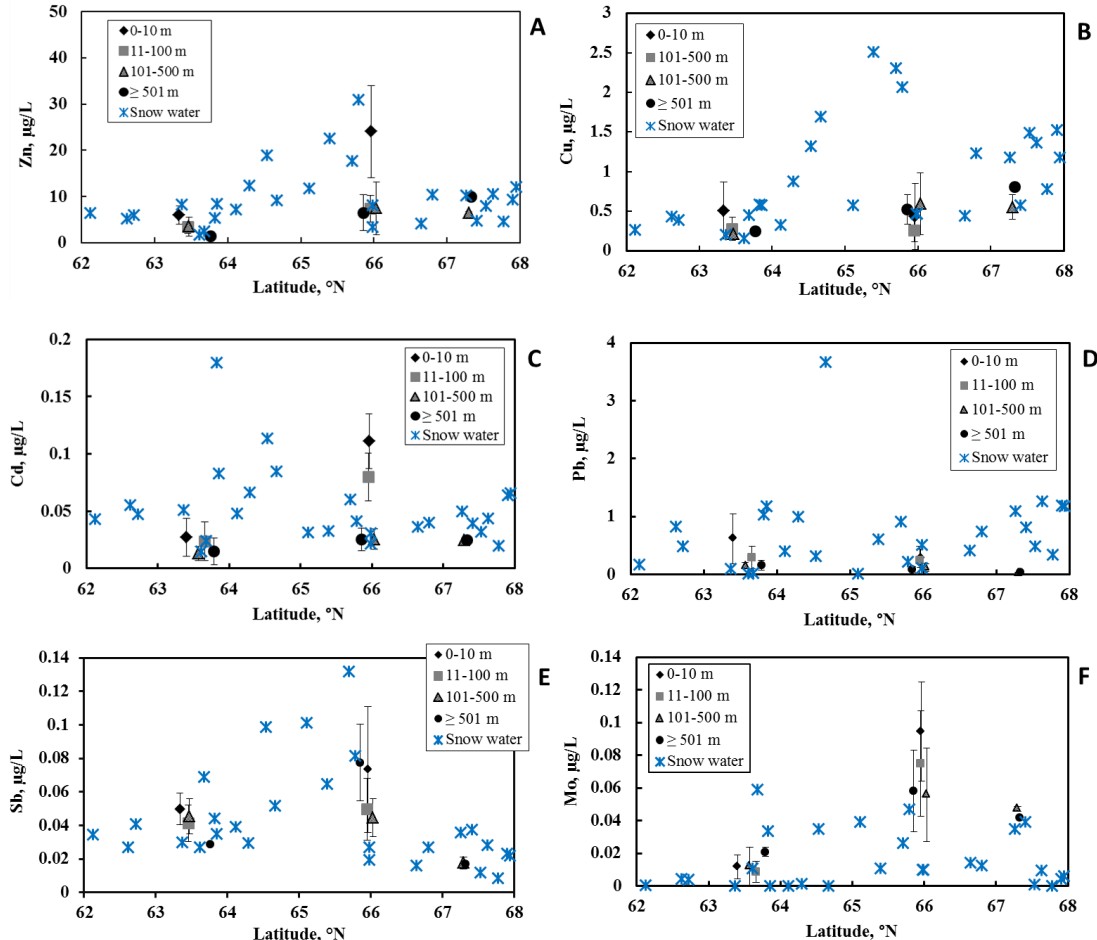


**Figure 5.** Snow water soluble (< 0.45 μm) concentration of Zn (A), Cu (B), Cd (C), Pb (D), Sb (E) and Mo (F) (blue
asterisk) compared with average concentrations in thermokarst lakes of different size in western Siberia (black symbols)
along the latitudinal gradient. Diamonds, squares, triangles and circles represent the lakes of four diameters: 0-10, 10-
100, 100 to 500, and > 500 m, respectively. The error bars represent the 2 s.d. of mean concentration for at least 10
lakes.







**Figure 6.** The ratio of the river water flux to that of the snow stock for elements that are most affected by atmospheric aerosols depositions. The flux ratio was calculated taking into account the snow volume (in mm of water) and river runoff (in mm during May and June).



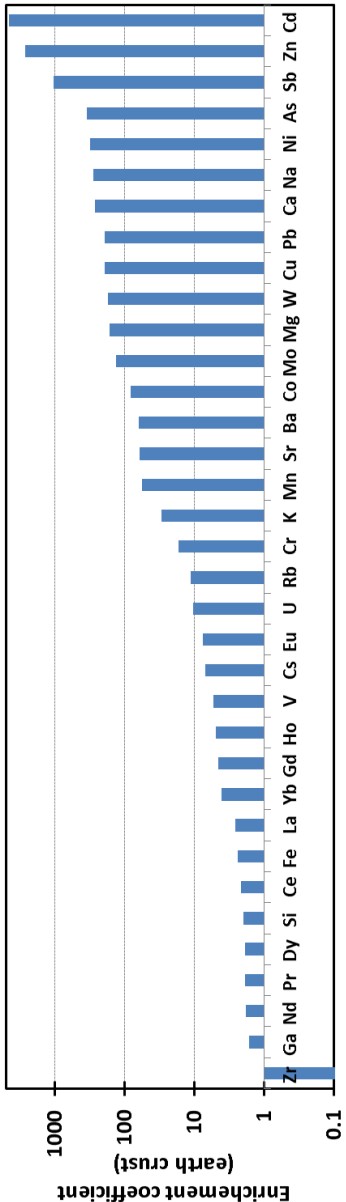

**Figure 7.** The latitude-averaged Al-normalized enrichment coefficient of snow particles with respect to the earth crust.





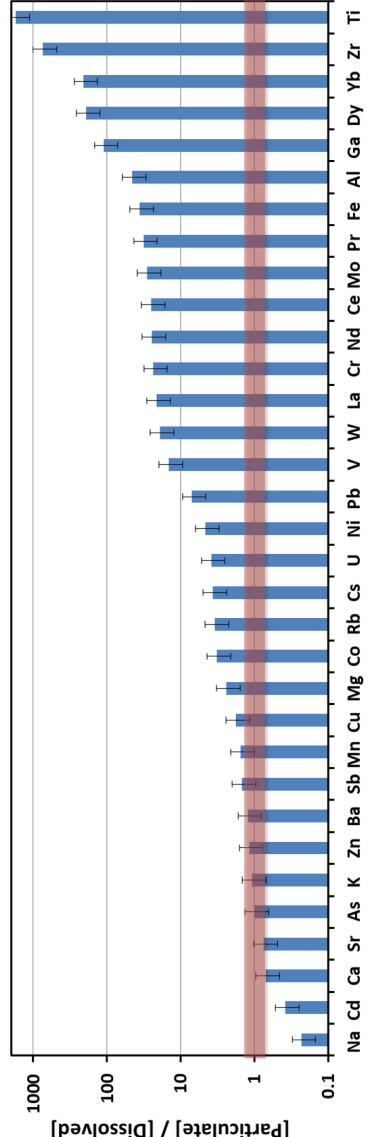

**Figure 8.** The average values (56 to 68°N) of the ratio of particulate to dissolved element concentration in snow water of western Siberia. Bold red line indicates statistically non-significant deviation from 1.





**Figure 9.** The ratios of the average concentrations of elements in snow particles (neglecting sample SF22) to those in mineral soil (A), peat (B) and mosses (C) of WSL. The peat, moss, and underlying mineral horizons data are averaged over the latitude of 55 to 68°N as described in Stepanova et al (2015). Note normal Y scale for mineral soil (A) and log Y scale for peat and moss (B, C).