# Peer review of "Impact of snow deposition on major and trace element"

_Hydrology and Earth System Sciences, 2016_

## Short Comment (SC1) · 9 Dec 2016

**REVIEW**

**Impact of snow deposition on major and trace element concentrations and fluxes in surface waters of Western Siberian Lowland**

**by Vladimir P. Shevchenko et al.**

**Introduction**

The article by Vladimir P. Shevachenko et al. titled "Impact of snow deposition on major and trace elements concentration and fluxes in surface waters of Western Siberian Lowland" attempts to contribute towards a better understanding of processes affecting element concentration and fluxes in the Western Siberian Lowland (WSL) by analyzing the snow deposition. Authors have looked at dissolved and particulate fraction of snow and compared it to this of river and lake waters. Then they compared the river fluxes in spring with snow water stock and investigated how the particles concentration and trace elements depended on snow particulate fraction. Additionally they tried to find a latitude influence. In the discussion they compared transportation fraction to find out that majority of measured elements was transported in particulate rather than dissolved fraction in the snow water. Then they investigated the effect of industrial and local pollutions and compared it to other places in the world. And finally they analyzed the atmospheric contribution to lakes and rivers composition. The main conclusions were 1) the particulate fraction of surface layer of snow "ranged from 0.4 to 66 mg/l of meltwater and increased in the regions of enhanced dust deposition." 2) the elementary composition of snow water dissolved fraction showed "a F1 x F2 structure with the first factor controlling the insoluble, low-mobile lithogenic elements and the 2nd factor acting on alkaline-earth metals, biogenic elements and anions." 3) the concentration of insoluble elements in snow water "did not change or decreased in response to the increase in PF." 4) the impact of the snowmelt on chemical composition of western Siberian thermokarst lakes and rivers may be very high it case of the rivers it increases northwards for several elements.

**Main assessment**

The article touches a fairly studied topic but in a area that has not yet been well documented what is being proclaimed by the limited data availability. The authors have done a lot of work and looked at the problem from many different aspects. I was impressed by the geographical area covered in this research as well as how many different aspects were looked at.
What more the article fits the scope of HESS and has a great potential to be interesting especially when put in the larger problematic such as climate change. On top of that it has created a significant amount of data and I get the impression that it will be very useful for future research in this area.

Nevertheless in the form it has now it appeared to me as not fully relevant. I could not sense the importance of the presented data and what do they mean for the Earth nor a smaller world of Siberia. The article's structure is somehow blur and it is difficult to follow the authors ideas. The sentences are often too long and therefore it is very easy to get confused. There are also language errors that make understanding what author wants to say difficult.

Research question and hypothesis seem too general. After a good introduction there is a hypothesis that suddenly tries to put the studied problem into a large scale of climate change but nowhere later is this topic developed.

To the large scale this project is based on the previous research which shows the topicality of the issue but also causes problems while reading because there are too many lines of references in the text. Many parts of the report contains the expression "described elsewhere" which in its frequency can be a bit irritating. The methods are already described in great detail and I do not see a reason for further information unless someone wants to based on this article but then they can contact the authors directly.

The graphics are not self-explanatory and could use more description otherwise without specific knowledge one cannot take the important message out of them. Also when the figure is adapted from previous studies I would state it in its description.

Despite these shortcomings, the data presented in this manuscript seems to be valuable and, thus, the publication of a significantly revised manuscript would be valuable.

**List of major and minor points**

**Title**
"Impact of snow deposition on major and trace elements concentration and fluxes in surface waters of Western Siberian Lowland"
I would recommend to specify what fluxes are being considered and add the information about latitude dependence analysis.

**Abstract**
I found it difficult to read and I had to read it several times before I understood it.

**Introduction**
In the introduction we see a clear link to previous studies. I would consider limiting the references. The references are listed but not given any explanation nor details.
Nevertheless the introduction clearly indicates the knowledge gap in the studies area and why the western Siberia was chosen.
What I find very good in the introduction is its direct statements on reasons of the originality of the conducted research.

**Method**
- All method used are presented to a great detailed that it would be possible to reproduce the data.
- I did not find a support for claim that "gas exploration facilities (..) minimally impact the environment."(page 6)
- It is good to state that the equipment was pre-cleaned but it is enough to do it once and not to focus so much on it afterwards.
- In the 2.4 there are results put in the methods part which I do not think belongs there (lines 160-164)
- When based on other results it is not indicated which method was used nor even stated whether it

was the same one or not.

- For someone who does not know how PCA works it is not explained well. I did not fully understand what does the F1 x F2 structure mean.

**Results**
- In the line 217-218 I would describe how can this effect be explained.
- In case of *Snow water in comparison to lake and river water* in line 238 it says that "only summer-period concentrations" were used. I would suggest to describe what possible consequences could it have.
- I liked the first general description followed by a regional example.
- When listing what elements increased (line 277) I would add information on what they have in common or a suggestion why they acted in the same way.
- The mineralogical composition was presented only for selected snow samples - it would be good to indicate what was the criteria for this selection.
- In the results the authors claim correlation while its ratio is less than 0.3. I would reconsider if that is not an effect of not enough data supply.

**Discussion and conclusions**
- The discussion part is missing more or less bold ideas of how do the authors explain the results. What do they think causes them (ex 334 - 335) and what do they mean for the bigger picture (ex. climate change) as mentioned in the introduction.
- Interesting results were obtained for latitude dependence and that is why I would add it to the title. I would also write authors thoughts on why in the figure 2 is there extreme at 64-66 for Ni, Cd and Sb.
- I am again missing the explanation on why is this important (ex 393 - 394)

**Graphics**
- The authors should improve the figures and especially the figure captions to make the figures easier to understand.
- Figures adopted from other studies should contain a proper reference.
- The statement that the Fig. S7 supports the claim that there is an increase in concentration north from 23-24*N does not seem to be the case on the presented figure.

**Language**
While reading the article I have seen many linguistic mistakes. (ex line 39, 48, 94, 110, 170-171, 492)

---

## Referee Comment (RC1) · Anonymous Referee #1 · 22 Dec 2016

General Comments

Shevchenko et al. proposes to show the impact of the snow chemical composition during snowmelt in the lake and river water of the western Siberia, a region of lowlands, which is still poorly studied. They acquired a very representative and complete dataset of chemical element concentrations on the snow of this region. By assessing the chemistry of both dissolved (+colloidal) and particulate fractions of snow, they address the effect of latitude and anthropogenic sources on major and trace element concentrations in snow as well as the interaction between dissolved (+colloidal) and

particulate snow fractions that are released in lake and river water of this region.

This manuscript corresponds to the scope of the journal, especially in presenting such a novel dataset in a permafrost area where geochemical and hydrological processes are still poorly studied. However, the submitted draft still needs significant improvement before to be published.

The presentation of the data should be done in a more synthetic way and not as a list of chemical elements, which change in each paragraph, making the reader completely lost after the first section of the result part. This comment should also be applied to the figures, which always present different chemical elements without any clear link between the different figures and also between one figure and its relative text. Keeping the reader interesting during all the manuscript when presenting such a large dataset is a real challenge and in my opinion this objective is not reached in this manuscript. Covering the large spectra of the ICP-MS analysis provides a more complete information on the snow, lake and river water composition but increases the difficulty to valorize the results with clarity, especially to an audience from various environmental disciplines. As the chemical elements do not behave equally in the different water type and in different locations this gives a difficulty to easily follow the description of the result, as they are presented, and the following discussions.

Proposing a working hypothesis could help to improve this presentation by given a clearer story-line to the manuscript. Then, the authors could choose a reduce number of elements to illustrate the defined hypothesis. Some parts of the discussions should be thoroughly addressed by improving the use of this amazing dataset. I see that the potential of the data is not fully exploited in regard to the discussions that are proposed in the manuscript.

Specific comments

Abstract

The abstract should be revised in order to clearly state the objective in regard to the existing knowledge on snow chemical composition at these latitudes. You should clearly explain the added value of this work to the international community.

Introduction

The introduction presents a clear state-of-the-Art on the knowledge of the chemical composition of snow under these latitudes but is lacking of a working hypothesis, which would be the directed line for the entire manuscript.

Study site, material and methods

This part is well structured and presents the important technical information to check the quality of this important dataset. You could nevertheless improve this section of the manuscript by removing some redundancies and merging some parts (citation of filtering in several parts of the methodology...). Calculation of EF could be added and explained according to recent literature in this part of the manuscript (see comment below).

Results

The results part is, in my opinion, the part that needs the main improvement. Using multivariate statistical analysis is a very good idea but is not fully exploited. Why not using the synthesis of such mathematical results to identify group of elements that behave in the same way and building the discussions on the elemental groups and not on the individual chemical element? But doing such an approach also requires to clearly stating the criteria that are used to identify the group of elements. I do not see the interest to propose such a classification in the present form of the manuscript and I do not understand the criteria that were used to define the different groups using this PCA analysis. For instance, how could the author justify that Cr and Zn are part of the same group that the other "lithogenic elements" (Fig. 3) and same question regarding elements of the second group. I think that a complementary hierarchical cluster

analysis should be done in order to clearly state the significance of a given element to be part of a given group that is shown in the PCA analysis of Fig. 3 and S5. (Kaufman and Rousseeuw, 2005. Finding Groups in Data. John Wiley and Sons Inc., NY). Such statistical technique has been widely used for drawing meaningful information from geochemical data (Bini et al., 2011 J. Geochemical Explor.; Levitan et al. 2015 J. Geochemical Explor.; Schot and van der Wal 1992 . Hydrol.; Gourdol et al, 2015, Appl. Geochem ; Moragues-Quiroga et al. 2017 Catena...). An interesting issue should be to compare the distribution of the elements in both snow fractions with the distribution of the same elements in the lake and river waters to observe some similarity/dissimilarity that could be explained by contrasted behaviour of elements after they are released from snow to river/lake water. Once the groups are significantly identified, you should replace the list of elements by the group number in order to clarify the results and discussion parts.

Some literature should be added to strengthen the similar behavior of elements that are present in a given group and then should behave in the same manner in the environmental fractions that are analyzed (snow, lake and river water, particles).

Using EF is always a debate, especially regarding the choice of the reference that has to be used for the normalization. How representative of the area geological characteristics is the chosen reference? How could this impact the results? Some publication should be considered to explain your choice like Reimann and de Caritat (2005 – Environ Sci Technol) and N'guessan et al. (2009 – Sci Tot Environ). This could be added to the methods by describing the EF calculation. As you used in the discussion soil/peat profiles collected in the WSL should you not relate EF to a more local geological/pedological information rather than an average Earth Crust?

Discussions

By filtering the snow and the lake/river water to 0.45$\mu$m, the dissolved fraction includes a colloidal load, which can play a crucial role in the concentration of trace elements.

Then, the <0.45$\mu$m dissolved fraction of the snow water (which may not include colloids, see lines 364-370) is compared to the <0.45$\mu$m dissolved fraction of river water (in which the trace element concentrations can be heavily controlled by colloids). If the colloids are not present in the dissolved fraction of snow, it is known that their contribution can be important in river, especially during flooding periods. Then the environmental compartment, which releases the colloids to the freshwater during the smelting period is not considered in the fluxes calculation of the present study and could overestimate the contribution of the dissolved snow fraction to freshwater fluxes. This point needs to be clarified in the discussions.

The part that discussed the anthropogenic impact is like a list of potential industrial and urban sources but no real proof regarding the exact origin of the analyzed snow particles is given. The use of elemental ratios could be interesting to use especially with such a dataset to relate with known and existing anthropogenic/natural sources in the investigated region.

Detailed comments

Line 39: ". . . of properties. . ."

Line 68: as the separation between dissolved and particulate fraction is <0.45$\mu$m, the dissolved one actually corresponds to a truly dissolved fraction associated to colloids. This colloidal load may be of importance in freshwater according to hydrological conditions and should be indicated in this manuscript when the dissolved fraction is described, especially if snow water is compared to lake and river water compositions. I propose to use dissolved + colloidal fraction

Line 76: correction "unprecedently"

Line 91: in contradiction to the discussion part on industrial impact on the snow particles

Line 137: indicate Rare Earth Element (REE)

Line 146: dissolved and colloidal

Line 286: what about Ga, Co and Cl, which present similar values in Fig. S4 than the element cited in the text?

Line 298: Mg and Ca are not plotted in Fig. 6.

Line 356: replace "lieu" by "origin or source..."

Line 362: please define LREE

L365: I agree regarding the marine aerosols but what about the atmospheric deposition that come from anthropogenic or lithogenic origins? Could they be enriched in Fe- or organic-colloids?

Line 377-379: "soil column" and "peat column" might not be the appropriate term, you could use "soil profile" and "peat profile" or "soil" and "peat".

Line 379: remove the symbol between "mosses" and "collected"

Lines 398-399: Is this not an important conclusion of this study? This should appears in the abstract and the conclusions. Figures A harmonization of the chemical elements presented in the different figures should be done for clarity improvement in the manuscript. Check the correspondence between the figures and the text: Ca and Mg do not appear in Figure 6... Figure 1: the different permafrost zones should be indicated Figure 3: caption should include more information on the parameterization of the PCA that was used. Figure S4: same scale should be used for the Y-axis of the three graphs.
* * *

---

## Author Comment (AC1) · 20 Jan 2017

Response to Anonymous reviewer No 2 (master student) We appreciate that the reader acknowledged that we ". . .have done a lot of work and looked at the problem from many different aspects". The reader "was impressed by the geographical area covered in this research as well as how many different aspects were looked at". We also acknowledge his/her opinion that ". . .the article fits the scope of HESS and has a great potential to be interesting especially when put in the larger problematic such as climate change".

However, the reviewer ". . . could not sense the importance of the presented data and

what do they mean for the Earth nor a smaller world of Siberia." In response to this remark, we would like to state that the snow deposition in Siberia is the main source of water which is transported to the Arctic Ocean by Siberian rivers, including dissolved and particulate elements. The importance of the riverine flux to the Arctic Ocean and the Arctic Ocean itself for the regulation of the climate and biota of the planet is well known and discussing this issue goes beyond the scope of this paper. We would also like to point out that, to the best of our knowledge, this is the first attempt to generalize the snow chemical composition across such a large latitudinal transect of the Arctic and subarctic zone.

We further agree with the reviewer that "...The article's structure is somehow blur and it is difficult to follow the authors ideas." In the revised version, we will clarify the presentation and will focus our description of results on most important findings.

The reviewer also noticed that "Research question and hypothesis seem too general". The working hypothesis for this study was that the chemical composition of snow should reflect the contribution of three main sources of dissolved and particulate forms of chemical elements. From the one hand, these are marine aerosols and mineral dust from remote deserts provinces, provided via long-range atmospheric transport. From the other hand, these are local sources of pollution such as towns, gas and oil industry centers and roads.

The reviewer correctly pointed out that "Many parts of the report contains the expression "described elsewhere" which in its frequency can be a bit irritating". We will carefully revise the description of methods and sites geography and remove the references to previous works as much as possible. Following the recommendation of this reviewer, we will revise the graphics to make them self-explanatory and we will provide more description, taking care of providing the proper reference to the previous figures. All the figures in the revised manuscript will be fully original.

The reviewer provided a list of detailed remarks; all of them are very useful and will be

carefully considered in the revised version. In particular, the cause for the latitudinal dependence in Fig 2 of elevated concentrations of Ni, Cd and Sb is most likely industrial impact, but given relatively low number of data points around industrial centers it is hard to prove it unambiguously. The main interest of this work was to provide a general view of element concentration across the WSL.

Further important remarks of reviewer were the request to provide the information on what the elements have in common or a suggestion why they acted in the same way (L 277). In fact, these are all soluble (highly labile) elements, originated either from marine aerosols or from leaching from soluble minerals such as carbonates. We avoided too much discussion in the Results section. The reviewer correctly pointed out that "The mineralogical composition was presented only for selected snow samples and it would be good to indicate what was the criteria for this selection". The choice was determined by sufficient amount of collected particulate phase for quantitative XRD and SEM analyses. These samples however, are highly representative for snow samples of Siberia and they cover full latitudinal range investigated in this study.

The reviewer also noticed that "In the results the authors claim correlation while its ratio is less than 0.3 and he/she wondered whether this is an effect of not enough data supply." This is certainly true. We reconsider the criterion of correlations using sophisticated multi-parametric statistics (PCA and newly added Hierarchical Cluster Analysis (HCA), see our response to reviewer No 1)

We thank the reviewer for generally positive opinion on our paper and we hope to have a chance of presenting the revised manuscript to the journal.

---

## Author Comment (AC2) · 20 Jan 2017

Response to anonymous Referee #1

The reviewer acknowledged "a very representative and complete dataset of chemical element concentrations on the snow of this region". He/she also considers that our manuscript "...corresponds to the scope of the journal, especially in presenting such a novel dataset in a permafrost area where geochemical and hydrological processes are still poorly studied."

However, the reviewer suggested to change the presentation of the data via i) distinguishing the groups of chemical elements and ii) providing a better link between the different figures and between each figure and its relative text. We will greatly revise the presentation following this recommendation. Note however, that for the first time we present both dissolved (< 0.45 $\mu$m) and particulate composition of snow across a huge latitudinal gradient. Given the complexity of various simultaneously acting factors on snow chemical composition (marine aerosols, remote deserts and local industrial pollution), and the lack of distinct geochemical signature of each individual source measured in western Siberia, establishing the links between different factors and between different group of elements become a challenge and goes beyond the scope of this work. We did distinguish several groups of elements using additional sophisticated statistical tools, suggested by reviewer (see below). This helped to focus the presentation and facilitated the discussion.

The reviewer suggests that "The presentation of the data should be done in a more synthetic way and not as a list of chemical elements". We agree and we will greatly revise the presentation focusing it on the groups of elements revealed by new statistics. For the dissolved part, these groups are 1) Pb, Al, Cr, REE; 2) Ti, Zr, V, As, Fe, Cu, Cd; 3) Na, Mg, Ca; 4) Cs, Mn, Rb, K, Co, Ba, and 5) Sr, Sb, Ni, Si, DOC, Mo which illustrate the combination of i) lithogenic silicate and refractory minerals subjected to leaching (1st and 2nd group), ii) soluble major ions (3rd group), and biogenic elements such as plant nutrients or highly labile components of the 4th group and 5th group. For the particulate fraction, new hierarchal cluster analysis revealed 5 groups of element depending on their leachability and the source (lithogenic silicates, local industry, marine aerosols).

The reviewer also suggested that "Proposing a working hypothesis could help to improve this presentation by given a clearer story-line to the manuscript." The working hypothesis for this study was that the chemical composition of snow should reflect the contribution of three main sources of dissolved and particulate forms of chemical elements. From the one hand, these are marine aerosols and mineral dust from

remote deserts provinces, provided via long-range atmospheric transport. From the other hand, these are local sources of pollution such as towns, gas and oil industry centers and roads. Formulating this hypothesis allowed to reduce the number of elements and discuss the group behavior.

The reviewer made a number of very constructive specific comments, among them: He/she stated that "The abstract should be revised in order to clearly state the objective in regard to the existing knowledge on snow chemical composition at these latitudes. You should clearly explain the added value of this work to the international community." This study provides for the first time, elementary composition of both dissolved and particulate fraction of snow over significant (1700-km) gradient of very important sub-arctic zone. We demonstrate significant and previously underestimated atmospheric input of many major and trace elements to their riverine fluxes during spring flood. The fundamental meaning of this finding is that the impact of snow deposition strongly increased northward, in discontinuous and continuous permafrost zones of frozen peat bogs, which is consistent with the decrease of the impact of rock lithology on river chemical composition in the permafrost zone of WSL, relative to the permafrost-free regions. A broader impact of this result is that current estimations of river water fluxes response to the climate warming in high latitudes may be unwarranted without detailed analysis of winter precipitation, which is largely ignored in most studies. Taking into account the snow deposition and its chemical composition, revealed for the first time for such a large arctic territory may change the dominant paradigm of increasing the nutrients and toxicants transport from the land to the ocean under climate warming in the Arctic.

The reviewer suggested improving the technical section by removing some redundancies and merging some parts. He/she also suggested that "Calculation of EF could be added and explained according to recent literature" and we carefully revised this part taking into consideration the recommended references of the group of Reimann and Probst. Here it is important to argue that, unlike the local pollution tracing in the Eu-
ropean arctic within the Kola Ecogeochemistry project (Reimann, de Caritat) or small-scale stream bed sediments or soils (N'guessan et al., 2009; Moragues-Quiroga et al., 2017; Levitan et al., 2015) where the normalization to the local soil or bedrock was necessary, the present study deals with winter-period long-range atmospheric transport of soluble and mineral forms of elements. As such, following the common practice in this field, normalization to general shale or Earth crust, allows representing the true enrichment/depletion of the atmospheric aerosols. Note that the normalization of the snow particles elementary composition to that of local mineral soil, peat and moss is presented in Fig. 9 of the manuscript and discussed in L 356-382 of submitted version.

The main criticism of this reviewer is on the results part. He/she suggested to "using the synthesis of such mathematical results to identify group of elements that behave in the same way and building the discussions on the elemental groups and not on the individual chemical element" using "a complementary hierarchical cluster analysis (Bini et al., 2011; Levitan et al., 2015; Schot and van der Wal, 1992; Gourdol et al., 2015; Moragues-Quiroga et al. 2017) to clearly state the significance of a given element to be part of a given group that is shown in the PCA analysis of Fig. 3 and S5. Following this important advice, we applied the methodology presented by Moragues-Quiroga (2017). Specifically, we conducted a hierarchical cluster analysis (e.g. Hartigan, 1975; Kaufman and Rousseeuw, 2005) which is widely adopted in geochemical interpretations of element concentration data (e.g. Bini et al., 2011; Levitan et al., 2015; Schot and van der Wal,1992). We used the Ward's method (Ward, 1963) for the linkages rule, following previous studies (Gourdol etal., 2013; Lin et al., 2014). The Pearson correlation distance was used for the linkage distance, which is frequently used for clustering variables (Reimann et al., 2008). These choices are in agreement with the group search of the PCA loadings. However, the obtained results of the HCA (not shown here) did not confirm the PCA maps. The group presentation on the correlation circle F1 x F2 did not allow fractionating the PCA level because the first two PCA factors present only a part of the data structure when it looks for the relationships between the variables (element concentration) and the samples whereas the HCA takes into account the full

set of data when it creates the groups as a function of the distances in the hyper-space without simplification of the structure by decomposition into the factors. Therefore, another analysis was conducted via constructing the data table on the basis of first two factors of the PCA. The HCA was applied on this new table which contained only the data structure described by two PCA factors. The results allowed partitioning of the variables. The criterion of non-intersection between the groups allowed to partition the chemical elements of the dissolved part into 6 groups presented in Figs 1 and 2 and the elements of the particulate fraction were distributed into 5 groups (Fig. 3 and 4). These groups characterize the elements according to their general chemical properties, their ability to mobilize in aqueous solution from the solid minerals, their affinity to the biota or their presence in the contaminated particles of industrial activity. Thus, the first two big groups of the dissolved fraction shown in Fig 1 and encircled in Fig. 2 comprise low mobile elements likely originated from alumino-silicate mineral matrix (Al, Cr, REE, Ti, Zr, Fe, V) as well as some volatile heavy metals typically present in mineral matrix (Cu, Cd, Pb). The 4th group includes major constituents of carbonate or marine aerosols matrix (elevated pH, Mg, Ca and Na). The 5th group includes typical macro- and micronutrients (K, Rb, Mn, Co, Ba). Finally, the last 6th group of elements comprises both labile elements linked to weatherable minerals (Sr, Sb, Si, Ni) and nutrients such as Sr, Ni, Si, DOC and Mo. Three of these elements are strongly enriched in snow particles relative to the Earth crust (Sr, Sb, Mo) thus suggesting their possible leaching from atmospheric dust into the soluble fraction of snow. We could not find a straightforward explication of the common group of Zn and U in soluble snow fraction (Fig. 1) For the particulate fraction, distinguishing the elements into 5 formal groups revealed by HCA shown in Fig. 3 and encircled in Fig. 4 is less certain and does not allow establishing clear link between the selected groups and physico-chemical properties of elements and their possible sources in the snow particles. Thus, in the 1st group, among three labile elements (Mg, Na and Ca) we identified V, which may exhibit elevated mobility in the form of anion in carbonate-bearing mineral particles. Divalent metals (Co, Ni, Mn) and Sr constitute the 2nd labile group of elements, yet this group also comprises Fe

and Cr, which should be rather associated with the 3rd group of insoluble low mobile elements marking the presence of phosphates (REE and P), refractory Zr and volatile Pb. The 4th group of elements revealed by HCA of particles is composed of Sb, Cu and Zn. All these elements are strongly enriched in snow particles over the soil minerals (see 7 and Fig. 9a of submitted manuscript). The last group of elements in snow particles comprise both labile (Li) and biologically-important Mo, K, Rb, Ba, toxic volatile elements which could bear the signature of anthropogenic pollution (As, Cd) but also low mobile Ti and Ga. We could not identify the link of elements in this group to the degree of snow particles enrichment relative to main "local" substrates of the WSL (moss, peat and clays), see below.

The reviewer further proposed an interesting issue, "...to compare the distribution of the elements in both snow fractions with the distribution of the same elements in the lake and river waters to observe some similarity/dissimilarity that could be explained by contrasted behavior of elements after they are released from snow to river/lake water." He/she argued that "Once the groups are significantly identified, we should replace the list of elements by the group number in order to clarify the results and discussion parts." We thank the reviewer for suggesting this excellent strategy of interpretation. The comparison of dissolved element concentration in lakes and snow demonstrated that for lakes, the snow loading clearly exceeds the concentrations in lakes of Pb, Zn, Cu, Cd, Sb and Mo (Fig. 5 of submitted manuscript). However, these elements belong to 4(!) various group of elements in dissolved snow fraction, identified by THE HCA (Fig. 1 of this response). In rivers, the dissolved elements that can be dominated by snow input are SO4, Cr, Co, Ni, Cu, Zn, Mo, Cd, Sb, Cs, W and Pb, which again, belong to 4 various group of elements identified by HCA.

It thus can be concluded that there is no direct link between the group of elements identified by cluster dendrogram in the snow water and the elements those concentration in rivers or lakes are significantly affected by snow deposition. We believe that a natural cause of this apparent inconsistency is different mechanisms controlling the

element distribution in the aerosols such as local sources of pollution, remote desert provinces, leaching of soluble elements from particulate fraction and in surface waters (interaction of melted snow with upper peat and moss/lichen horizons; underground feeding, leaching of elements from silicate river suspended matter due to abrasion in spring flood).

The reviewer pointed out that "some literature should be added to strengthen the similar behavior of elements that are present in a given group and then should behave in the same manner in the environmental fractions that are analyzed (snow, lake and river water, particles)". We thank the reviewer for pointing this out and we greatly extended the reference list (see References of this reply). There are a few elements that behave similarly in lakes, rivers and snow water (see our response to the previous comment). We would like to underline that the geochemical behavior (migration, correlation, degree of enrichment) of elements is fundamentally different in rivers, lakes and winter atmospheric aerosols. Comparative analysis of these processes is already considered in our previous publications on geochemistry of surface waters in western Siberia (Audry et al., 2011; Pokrovsky et al., 2011, 2013, 2014, 2015, 2016a,b; Shirokova et al., 2013) goes beyond the scope of the present manuscript.

The reviewer stated that "using EF is always a debate, especially regarding the choice of the reference that has to be used for the normalization. How representative of the area geological characteristics is the chosen reference? How could this impact the results? Some publication should be considered to explain your choice like Reimann and de Caritat (2005 – Environ Sci Technol) and N'guessan et al. (2009 – Sci Tot Environ). This could be added to the methods by describing the EF calculation". We agree and will do so in the revised version. See our response to general comment above. We actually use 4 references for the area geological characteristics – average Earth crust, "local" moss, peat and underlying mineral horizon. The use of average crust is justified by long-range transfer of snow components. It is known since the works of group of Reimann and de Caritat in NW Europe that the "average crust" is unlikely

to represent the local background and the use of the "upper crust" average value can introduce a 2 to 3 order of magnitude uncertainty to any calculated EF (de Caritat et al., 1997; Reimann and de Caritat, 2000; Reimann et al., 2000). As such, western Siberia moss, peat and clay/loam horizons were used to assess relative enrichment of elements in snow particles. It can be assumed that the leaching of soluble forms of elements from these solid phases in winter is highly unlikely. The specificity of western Siberia is that the mineral ("geological") local substrate is completely frozen, even in summer, since the active (unfrozen) layer depth does not exceed the peat thickness, and in that case, the use of "organic" substrates is most relevant. All three WSL reference substances ("local"moss, peat and clays) represent latitudinal-averaged values based on large (> 50) number of samples collected in previous studies across the 1700-km latitudinal gradient.

He/she further inquired "As you used in the discussion soil/peat profiles collected in the WSL should you not relate EF to a more local geological/pedological information rather than an average Earth Crust? This is thoroughly discussed in L 356-382 and illustrated in Fig. 9 of submitted manuscript. It is hard to provide more "local" geological and pedological reference than the moss, peat and underlying mineral deposits sampled along a similar transect. To our knowledge, such a thorough comparison of snow to the dominant local geogenic background has never been attempted in the arctic region over a large latitudinal profile.

Comments of reviewer on Discussion: "By filtering the snow and the lake/river water to 0.45 $\mu$m, the dissolved fraction includes a colloidal load, which can play a crucial role in the concentration of trace elements. Then, the <0.45 $\mu$m dissolved fraction of the snow water (which may not include colloids, see lines 364-370) is compared to the <0.45 $\mu$m dissolved fraction of river water (in which the trace element concentrations can be heavily controlled by colloids). If the colloids are not present in the dissolved fraction of snow, it is known that their contribution can be important in river, especially during flooding periods. Then the environmental compartment, which releases the colloids

to the freshwater during the melting period is not considered in the fluxes calculation of the present study and could overestimate the contribution of the dissolved snow fraction to freshwater fluxes. This point needs to be clarified in the discussions." We totally agree that the generation of colloids during springflood is mostly linked to DOC mobilization from organic topsoil and plant litter. We would like to point out that the < 0.45 $\mu$m fraction of snow includes the colloids (1 kDa – 0.45 $\mu$m) by definition. However, with typical concentration of DOC in snow water around 1-2 mg/L, the share of colloidal forms of metals will be an order of magnitude lower than that in river and lakes, having 10 to 30 mg/L of DOC. A thorough discussion of colloids in surface waters of western Siberia is presented in our recent work (Pokrovsky et al., 2016, Geochim. Cosmochim. Acta) and thus we avoided too extensive discussion in the present manuscript.

The part that discussed the anthropogenic impact is like a list of potential industrial and urban sources but no real proof regarding the exact origin of the analyzed snow particles is given. The use of elemental ratios could be interesting to use especially with such a dataset to relate with known and existing anthropogenic/natural sources in the investigated region. This point is well taken. In agreement with the bulk of available information on metal pollutants in atmospheric precipitates (works of Reimann and de Caritat in the Kola Peninsula), we do agree that the anthropogenic pollution cannot be evidenced by mere enrichment factors. Moreover, we do not interpret the elevated concentrations of divalent metals, As and Sb as necessarily pollution from the industrial centers. Rather, volatile Pb, Cd, As may originate from long-range transport of desert material. We do discuss the fly ash (burning spheres) distribution in the context of gas flaring and road pollution (Section 4.2). Following recommendation of reviewer, we attempted to distinguish the well-known refractory, non-volatile heavy metals such as Cu, Ni and Co and more volatile elements such as Pb, Cd and As (i.e., Reimann et al., 2000) based on the new HCA treatment. For both particulate and dissolved fraction, these elements are located in three or two different groups but never belong to one single group of inter-correlated elements. As such the available data do not evidence similar origin of Cu, Ni and Co, or Pb, Cd, and As in the snapshot of WSL

snow sampled in this work.

The reviewer also provided a list of very useful detailed comments, most important of them are addressed below:

Line 286: what about Ga, Co and Cl, which present similar values in Fig. S4 than the element cited in the text? The riverine spring-time fluxes of Ga, Co and Cl in the southern, permafrost –free zone of the WSL can be provided by melted snow.

Line 298: Mg and Ca are not plotted in Fig. 6. Fig. 6 presents only the elements those fluxes in rivers are affected by snow deposition. See Fig. S4 of Supplement for Mg and Ca fluxes.

L365: I agree regarding the marine aerosols but what about the atmospheric deposition that come from anthropogenic or lithogenic origins? Could they be enriched in Fe- or organic-colloids? There is no known anthropogenic source of Fe north of 63°N. The lithogenic Fe from underlying clays is also unlikely since the mineral (clay, silt) horizons is permanently frozen in the continuous permafrost zone of the WSL.

Lines 398-399: Is this not an important conclusion of this study? This should appears in the abstract and the conclusions. We totally agree. The supply of mineral particles from the snow may also significantly enrich the rivers and lakes in dissolved alkaline earths, metal micronutrients, phosphorus and other elements given high reactivity of incoming silicate and carbonate grains in acidic (pH < 3-4), organic-rich (10 < DOC < 50 mg/L) surface waters of Western Siberia. Note that we currently collecting the dust from the Kazakhstan steppe for laboratory leaching by western Siberian surface waters.

Check the correspondence between the figures and the text: Ca and Mg do not appear in Figure 6. This is true. Shown in this figure are only the elements that are sizably affected by atmospheric deposition, averaged across full WSL territory. Ca and Mg appear only in Fig S4 C, for the continuous permafrost zone which shows indeed, the
similarity of snow deposition of Ca and Mg and the flux of these elements in rivers. In the revised version, we would like to place the Figure S4 in the main text.

References Bini, C., Sartori, G.,Wahsha, M., Fontana, S., 2011. Background levels of trace elements and soil geochemistry at regional level in NE Italy. J. Geochemical Explor. 109, 125–133. http://dx.doi.org/10.1016/j.gexplo.2010.07.008. de Caritat, P., Reimann, C., Chekushin, C., Bogatyrev, I., Niskavaara, H., Braun, J., 1997. Mass balance between emission and deposition of airborne contaminants. Environ. Sci. Technol. 31, 2966-2972. Gourdol, L., Hissler, C., Hoffmann, L., Pfister, L., 2013. On the potential for the Partial Triadic Analysis to grasp the spatio-temporal variability of groundwater hydrochemistry. Appl. Geochem. 39, 93–107. http://dx.doi.org/10.1016/j.apgeochem.2013.10.002. Hartigan, J., 1975. Clustering Algorithms. John Wiley and Sons, NY. Kaufman, L., Rousseeuw, P.J., 2005. Finding Groups in Data. JohnWiley and Sons Inc., NY (368pp). Levitan, D.M., Zipper, C.E., Donovan, P., Schreiber,M.E., Seal, R.R., Engle,M.a., Chermak, J.a., Bodnar, R.J., Johnson, D.K., Aylor, J.G., 2015. Statistical analysis of soil geochemical data to identify pathfinders associated with mineral deposits: an example from the Coles Hill uranium deposit, Virginia. USA. J. Geochemical Explor. 154, 238–251. http://dx.doi.org/10.1016/j.gexplo.2014.12.012. Lin, X.,Wang, X., Zhang, B., Yao, W., 2014. Multivariate analysis of regolith sediment geochemical data from the Jinwozi gold field, north-western China. J. Geochemical Explor. 137, 48–54. http://dx.doi.org/10.1016/j.gexplo.2013.11.006. Moragues-Quiroga C.„ Juilleret J., Gourdol L., Pelt E., Perrone T., Aubert A., Morvan G., Chabaux F., Legout A., Stille P., Hissler C. 2017. Genesis and evolution of regoliths: Evidence from trace and major elements and Sr-Nd-Pb-U isotopes. Catena 149 (2017) 185–198 Pokrovsky O.S., Manasypov R.M., Loiko S.V., Shirokova L.S., 2016b. Organic and organo-mineral colloids of discontinuous permafrost zone. Geochimica Cosmochimica Acta, 188, 1-20. Reimann, C., Filzmoser, P., Garrett, R.G., Dutter, R., 2008. Statistical Data Analysis Explained: Applied Environmental Statistics with R. John Wiley and Sons Ltd., NY (343pp). Reimann, C., de Caritat, P., 2000. Intrinsic flaws of element enrichment fac-

tors (EFs) in environmental geochemistry. Environ. Sci. Technol. 34, 5084-5091. Reimann, C., Banks, D., de Caritat, P., 2000. Impacts of airborne contamination on regional soil and water quality: The Rola Peninsula, Russia. Environ. Sci. Technol., 34, 2727-2732. Schot, P.P., van der Wal, J., 1992. Human impact on regional groundwater composition through intervention in natural flow patterns and changes in land use. J. Hydrol. 134, 297–313. http://dx.doi.org/10.1016/0022-1694(92)90040-3.

[Figure]
Interactive
comment

**Cluster Dendrogram**

dissolved fraction

**Figure 1**. Dendrogram of a hierarchical cluster performed on variables of a reconstructed table for the dissolved fraction using Pearson correlation distance as distance measure and Ward's method for the linkage rule.

**Fig. 1.**

[Figure]

dissolved fraction

**Figure 2.** PCA Factorial map F1xF2 of elements of a reconstructed table for the dissolved fraction. Partition of elements into 6 groups revealed by a CAH is shown by a contour line.

**Fig. 2.**

[Figure]

**Figure 3.** Dendrogram of a hierarchical cluster performed on variables of a reconstructed table for the particulate fraction using Pearson correlation distance as distance measure and Ward's method for the linkage rule.

**Fig. 3.**

[Figure]

particulate fraction

**Figure 4.** PCA Factorial map F1xF2 of variables (elements) of a reconstructed table for the particulate fraction. Partition of elements into 5 groups revealed by a CAH is reported by a contour line.

**Fig. 4.**

---

## Referee Comment (RC2) · Anonymous Referee #2 · 8 May 2017

General suggestions:

There is very limited knowledge of chemical and particulate composition of snow and water in Western Siberia currently. This manuscript shows a complete dataset of chemical element contents on the snow through sampling of substantial latitudinal transect in this region. The authors addressed the latitudinal effect, particle mineralogical impact and contribution separation of river input and atmospheric deposition to the lake. The topic and scope of this paper is a good fit for the journal. However, the current version of the manuscript confronted the following issues in my opinion, so that I suggest the

manuscript be major revised.

Specific suggestions:

Page 1, Line 15-31, The Abstract section need a clear objective (at the beginning) as well as significance (at the end). Result descriptions should summarily focus on the objectives.

Significant editing is required to improve the grammar, syntax and English expression throughout the paper. See comments on page 2 for examples:

Page 2, Line 39, "exhibits a number of properties"

Page 2, Line 46, "the chemistry of winter atmosphere" could be changed to "the air chemistry in winter"?

Page 2, Line 48, "atmospheric transfer" changed to "atmospheric transport"?

Page 2, Line 76; Page 3, Line, 89: do not often use the words like "unprecedentedly", "exceptionally", etc.

Page 3, Line 105: "we chose to sample..."

Page 3, Line 110: "All sampling points were located more than 500 m far from the winter road."

Page 4, Line 134: as "trace elements" was defined as TE, it should be consistent to use TE through the manuscript. Please check it. In addition, need to define REEs at Line 136.

Page 5 Line 171: "...performed, taking into account of ..."

Page 5 Line 173: "...calculated from hydrological parameters."

Page 5 Line 178-180: this is a bad sentence, "The most recent complete hydrological data of small and medium size rivers in permafrost-affected area of WSL (Novikov et al., 2009) were used together with RHS database to calculate the spring flood fluxes

of individual rivers and snow water stock for three latitudinal zones. . ..”

The second paragraph of the Introduction (Line 51-65) includes a lot of information and previous literature review. This could be improved by re-organizing or combining some of the previous reports in different regions. Line 51: "numerous studies . . ..western Siberia", probably in contradiction to the text on Line 61: "the trace . . . Siberian snow remains at the beginning of exploration."

Results: the database was not clearly and continuously expressed. Too many small figures and supplement information. Readers have to obtain the data information here and there. Although it is difficult to think out a reasonable method to improve the description of big database, the authors still need to consider it.

Page 6, Line 209-213: the PCA results seems inexplicable. Fe and Pb were attributed to lithogenic stable group; however, Ba and Si to highly mobile group. In the lithogenic group, Ti was not observed. If the mobile elements were used as indicator of marine aerosols, the element, Na should be also taken into account.

Page 8, Line 299-306: The introduction of EF calculation should be moved to the Method section. The authors should give some reasons to choose Al to be reference element, rather than Si or Ti in this study. Is there a good relation between Al and other trace elements? In addition, it is interesting to see that the EFs of Pb, Cu and Sb are larger than 100, and those of Sb, Zn and Cd larger than 1000, which clearly show serious anthropogenic pollution. However, in the introduction, the authors said that the region experiences lower anthropogenic disturbance, with fewer people and less industry. So, how to explain the high EFs? Probably it is a result of using average Earth Crust as the background?

Discussion: The authors pointed out three objectives: latitudinal effect, mineralogical impact and different sources of metal input to lakes. So, the discussions should be related to these key issues. The current version of this section should be shorten, as some of text repeated the Results content.

[Figure]

Conclusions: This section should be compressed. It is not necessary to repeat too much details of data results.

Quality/resolution could be improved for all figures. Figures 6, 7 and 8 could be merged together.
* * *

---

## Author Comment (AC4) · 15 May 2017

The reviewer noted that "The topic and scope of this paper is a good fit for the journal." However, he/she suggested major revision of the manuscript via addressing the following issues. Specific suggestions of Reviewer No 2:

- Reviewer comment: Page 1, Line 15-31, The Abstract section needs a clear objective (at the beginning) as well as significance (at the end). Result descriptions should summarily focus on the objectives. - Author reply: We revised the Abstract as following: "Towards a better understanding of chemical composition of snow and its impact

on surface water hydrochemistry in poorly studied Western Siberia Lowland (WSL), the surface layer of snow was sampled in February 2014 across a 1700-km latitudinal gradient (c.a. 56.5 to 68°N) in essentially pristine regions. We aimed at assessing the latitudinal effect on both dissolved and particulate forms of element in snow in order to quantify the possible source of atmospheric input to lake water solutes and elementary fluxes of rivers across the WSL. The concentration of dissolved+colloidal (< 0.45 $\mu$m) Fe, Co, Cu, As, La increased by a factor of 2 to 5 north of 63°N compared to southern regions. The pH and dissolved Ca, Mg, Sr, Mo and U in snow water increased with the increase in concentration of particulate fraction (PF). Principal Component Analyses of major and trace element concentration in both dissolved and particulate fractions revealed 2 factors not linked to the latitude. A hierarchical cluster analysis yielded several group of elements originated from alumino-silicate mineral matrix, carbonate minerals and marine aerosols or belonging to volatile atmospheric heavy metals, labile elements from weatherable minerals and nutrients. The main sources of mineral components in PF are desert and semi-desert regions of central Asia. Comparison of major and trace elements in dissolved fraction of snow with lakes and rivers of western Siberia across the latitudinal gradient revealed significant atmospheric input of a number of trace elements to the inland waters of the WSL. The snow water concentration of DIC, Cl, SO4, Mg, Ca, Cr, Co, Ni, Cu, Mo, Cd, Sb, Cs, W, Pb and U exceeded or were comparable with spring-time concentration in thermokarst lakes of the permafrost-affected WSL zone. The spring-time river fluxes of DIC, Cl, SO4, Na, Mg, Ca, Rb, Cs, metals (Cr, Co, Ni, Cu, Zn, Cd, Pb), metalloids (As, Sb), Mo and U in the discontinuous to continuous permafrost zone (64-68°N) can be explained solely by melting of accumulated snow. The impact of snow deposition on riverine fluxes of elements strongly increased northward, in discontinuous and continuous permafrost zones of frozen peat bogs. This was consistent with the decrease of the impact of rock lithology on river chemical composition in the permafrost zone of WSL, relative to the permafrost-free regions. Therefore, the present study demonstrates significant and previously underestimated atmospheric input of many major and trace elements to their riverine fluxes during spring flood. A

broader impact of this result is that current estimations of river water fluxes response to the climate warming in high latitudes may be unwarranted without detailed analysis of winter precipitation."

- Reviewer comment: Significant editing is required to improve the grammar, syntax and English expression throughout the paper. - Author reply: The manuscript already received full English proofread via payed service at the Elsevier Webshop. We corrected a number of syntax and English expressions errors in the text following valuable recommendations of 1st reviewer, the Master Student review and Reviewer No 2. We performed all recommended corrections to L 39, 46, 48, 76, 105, 110, 134, 136, 171, 173, 178-180 etc. Note that in case of acceptance of our paper for HESS, it will received full English proofread as a part of payed Open access package.

- Reviewer comment: The second paragraph of the Introduction (Line 51-65) includes a lot of information and previous literature review. This could be improved by re-organizing or combining some of the previous reports in different regions. - Author reply: We greatly shortened and reorganized the Introduction (see revised ms as attachement)

- Reviewer comment: Line 51: "numerous studies ...western Siberia", probably in contradiction to the text on Line 61: "the trace ...Siberian snow remains at the beginning of exploration." - Author reply: We removed this line of text as unnecessary.

- Reviewer comment: Results: the database was not clearly and continuously expressed. Too many small figures and supplement information. Readers have to obtain the data information here and there. Although it is difficult to think out a reasonable method to improve the description of big database, the authors still need to consider it. - Author reply: We strongly agree with this comment. We completely re-organized the results presentation, renumbered all figures and tables. The Results now follow the plan: 1) soluble fraction of element in snow water; 2) chemical composition of the particulate fraction and 3) assessing the impact of snow deposition on chemistry of lakes

and rivers, see attached manuscript. We would like to point out that it is inevitable to have small figures for representing individual elements. It is not always possible to combine several elements on the same plots. However, following this recommendation, we removed significant number of figures from Supplement (altogether more than 12 plots) which allowed greatly simplifying the presentation of results.

- Reviewer comment: Page 6, Line 209-213: the PCA results seem inexplicable. Fe and Pb were attributed to lithogenic stable group; however, Ba and Si to highly mobile group. In the lithogenic group, Ti was not observed. - Author reply: The reviewer made a very good point. The PCA was not sufficiently powerful to explain the variability of major and trace elements in both dissolved and particulate phase of snow water. To address this issue, we used alternative technique to identify the group of elements that behaved in a similar way in snow water and snow particles. For this, we applied a complementary hierarchical cluster analysis (HCA) (e.g. Hartigan, 1975; Kaufman and Rousseeuw, 2005) which is widely adopted in geochemical interpretations of element concentration data (e.g. Bini et al., 2011; Levitan et al., 2015; Schot and van der Wal,1992; Moragues-Quiroga (2017). We used the Ward's method (Ward, 1963) for the linkages rule, following previous studies (Gourdol etal., 2013; Lin et al., 2014). The Pearson correlation distance was used for the linkage distance, which is frequently used for cluster variables (Reimann et al., 2008). These choices are in agreement with the group search of the PCA loadings. The HCA analysis was conducted on the basis of first two factors of the PCA. The criterion of non-intersection between the groups allowed partitioning the chemical elements of the dissolved part into 6 specific groups presented in Fig. 1 B. These groups characterize the elements according to their general chemical properties, ability to mobilize in aqueous solution from the solid minerals, affinity to the biota or their presence in the contaminated particles of industrial activity. Thus, the first two group of the dissolved fraction shown in Fig. 1 A and encircled in Fig. 1 B comprise low mobile elements likely originated from alumino-silicate mineral matrix (Al, Cr, REE, Ti, Zr, Fe, V) as well as some volatile heavy metals typically present in the solid aerosol particles (Cu, Cd, Pb). The 4th group includes

major constituents of carbonate or marine aerosols matrix (elevated pH, Mg, Ca and Na). The 5th group is represented by typical macro- and micronutrients (K, Rb, Mn, Co, Ba). Finally, the last 6th group of elements comprises both labile elements linked to weatherable minerals (Sr, Sb, Si, Ni) and nutrients such as Sr, Ni, Si, DOC and Mo. Three of these elements are strongly enriched in snow particles relative to the Earth crust (Sr, Sb, Mo, see section 3.3 of revised manuscript), thus suggesting their possible leaching from atmospheric dust into the soluble fraction of snow. We could not find a straightforward explication of the common group of Zn and U in soluble snow fraction.

- Reviewer comment: If the mobile elements were used as indicator of marine aerosols, the element, Na should be also taken into account. - Author Reply: The PCA identified the second large group which was composed of DOC, K, Rb, Cs, Mn, Co, Ba, Sb, Co, Mo, Mg, Si, Sr, Na, Ca, pH. These highly mobile elements presumably reflect the marine aerosols and leaching from soluble soil minerals such as carbonates as well as plant biomass. In the HCA treatment, Na is located in the same group with Mg and Ca, the two major elements of marine aerosols (see Fig. 1 B).

- Reviewer comment: Page 8, Line 299-306: The introduction of EF calculation should be moved to the Method section. The authors should give some reasons to choose Al to be reference element, rather than Si or Ti in this study. Is there a good relation between Al and other trace elements? - Author Reply: Yes, there is a very good linear relation between Al and other immobile trace elements such as Ga, Zr, Th, Ti shown in Fig. 2 of Reply. The use of Al-normalized TE enrichment factor (EF) was for consistency with all previous results on snow particles composition in the Russian Arctic and subarctic: normalization to Al is the most common way of data presentation.

- Reviewer comment: In addition, it is interesting to see that the EFs of Pb, Cu and Sb are larger than 100, and those of Sb, Zn and Cd larger than 1000, which clearly show serious anthropogenic pollution. However, in the introduction, the authors said that the region experiences lower anthropogenic disturbance, with fewer people and less industry. So, how to explain the high EFs? Probably it is a result of using average Earth

Crust as the background? - Author reply: Here, one has to distinguish the local pollution or anthropogenic disturbance that are known to produce the enrichment of atmospheric precipitates in toxic metals, for instance, in the vicinity (50-100 km) of Arctic smelters (works of de Caritatn, Reinmann), and long-range atmospheric pollution. We believe that the elevated concentrations of divalent metals, As and Sb should not be interpreted as necessarily pollution from the industrial centers. Rather, volatile Pb, Cd, As may originate from long-range transport of anthropogenic pollutants and the desert material. Therefore, we attempted to distinguish the well-known refractory, non-volatile heavy metals such as Cu, Ni and Co and more volatile elements such as Pb, Cd and As (i.e., Reimann et al., 2000) based on the HCA treatment. For both particulate and dissolved fraction, these elements are located in three or two different groups but never belong to one single group of inter-correlated elements. As such the available data do not evidence similar origin of Cu, Ni and Co, or Pb, Cd, and As in the snapshot of WSL snow sampled in this work. To further address this comment, we compared the element in snow samples to their concentration in local geological background rather than average earth crust (see Fig. 3 of Reply). Although the use of average crust for assessment of element enrichment in snow particles is justified by long-range transfer of snow components, it is known since the works of group of Reimann and de Caritat in NW Europe that the "average crust" is unlikely to represent the local background and the use of the "upper crust" average value can introduce a 2 to 3 order of magnitude uncertainty to any calculated EF (de Caritat et al., 1997; Reimann and de Caritat, 2000; Reimann et al., 2000). As such, western Siberia moss, peat and clay/loam horizons were used to assess relative enrichment of elements in snow particles. It can be assumed that the leaching of soluble forms of elements from these solid phases in winter is highly unlikely. The specificity of western Siberia is that the mineral ("geological") local substrate is completely frozen, even in summer, since the active (unfrozen) layer depth does not exceed the peat thickness, and in that case, the use of "organic" substrates is most relevant. All three WSL reference substances ("local" moss, peat and clays) represent latitudinal-averaged values based on large (>

50) number of samples collected in previous studies across the 1700-km latitudinal gradient. The elementary ratios of snow particles to that in mineral soil, peat and moss of the WSL are illustrated in Fig. 3 A, B, and C, respectively.

- Reviewer comment: Discussion: The authors pointed out three objectives: latitudinal effect, mineralogical impact and different sources of metal input to lakes. So, the discussions should be related to these key issues. - Author reply: We totally agree with this remark. The revised discussion is constructed as following: 1) Dissolved fraction of snow, comparison with literature data and discussion of the origin of elements; 2) particulate transport of elements in snow, concentration of particulate fraction, its origin, and 3) Impact of snow deposition on river and lake chemistry and fluxes

- Reviewer comment: The current version of this section should be shorten, as some of text repeated the Results content. - Author reply: We agree and greatly revised this section via removing 20 lines of text, see the revised manuscript.

- Reviewer comment: Conclusions: This section should be compressed. It is not necessary to repeat too many details of data results. - Author reply: We agree and removed 13 lines of text from the Conclusions

- Reviewer comment: Quality/resolution could be improved for all figures. Figures 6, 7 and 8 could be merged together - Author reply: We strongly revised many figures and removed part of figures from the manuscript (see attachment)

The revised manuscript, which incorporates all the comments of MSc, Reviewer 1, and Reviewer 2, is attached as Supplement to this Reply.

Please also note the supplement to this comment:
http://www.hydrol-earth-syst-sci-discuss.net/hess-2016-578/hess-2016-578-AC4-supplement.pdf
* * *
[Figure]

[Figure]

**Figure 1 Reply:** PCA Factorial map F1xF2 of elements of a reconstructed table for the dissolved fraction. Partition of elements into 6 groups revealed by a CAH is shown by a contour line. **B:** Dendrogram of a hierarchical cluster performed on variables of a reconstructed table for the dissolved fraction using the Pearson correlation as a distance measure and Ward's method for the linkage rule.

**Fig. 1.**

**Figure 2**. Relationship between Al concentration in the particulate fraction and that of Ti, Zr, Ga and Th

**Fig. 2.**

**A**

SNOW / mineral soil

Ti K Zr Hf Th Cs Rb Nd Ce Pr Yb U La Ga Y Al Tl Fe V Na Sr Ba P Mn Co Cr Ge As Ca Cu Mo Pb Mg Cd Ni Zn Sb

**B**

SNOW / PEAT

Ge P Cd Ti Sr U Mo Fe Hf Th Nd Ce Pr Ca Yb La Y Zr Cs Rb Tl K Mn Co Al Ga Ba V As Na Zn Cu Sb Pb Cr Ni Mg

**C**

SNOW / MOSS

P Cd K Rb Mn Cs Tl Ge Hf Ti Pb Sr Mo Zn Ca Fe As U Sb Nd Ce Pr Yb Zr Ga La Y Ba Al Na V Th Cu Co Mg Cr Ni

**Figure 3.** The ratios of the average concentrations of elements in snow particles (neglecting sample SF22) to those in mineral soil (A), peat (B) and mosses (C) of WSL. The peat, moss, and underlying mineral horizons data are averaged over the latitude of 55 to 68°N as described in Stepanova et al (2015). Note normal Y scale for mineral soil (A) and log Y scale for peat and moss (B, C).

**Fig. 3.**

---

## Author Response (AR2)

Dear Dr Laurent Pfister,

After second round of review of our revised manuscript, the reviewer stated that scientific significance, quality and presentation of revised manuscript are at the good level and recommended publication pending several minor corrections.

The comments of reviewer were addressed as following:

- *Lines 212-218 should be put in the methods to indicate the potential blank contributions. You should also provide the metal content of the acetate cellulose filters that are digested with the sample to estimate the elemental particulate concentrations in snow.* We agree and shifted this part to the Methods (now L 158-165). Note that these are MilliQ blanks of soluble fraction of snow. The second remark is on the blanks of filter digestion. We addressed this issue in submitted ms (L 127-129) and revised the text as following: "For the analysis of snow particles on filters, the blanks were estimated after digestion of 6 random filters. In the digestion solution, the concentrations of all trace elements were a factor of 10 to 100 lower than that obtained from the filters with particles after 0.5-1.0 L of snow water filtration", L 127-130. We do not think that adding a table with blank analysis will be necessary and we do not have sufficient statistics to recommend the values of metal concentration in commercial (Millipore) acetate cellulose filters. However, we added, as supplement to this reply, a compilation of average major and TE concentration in digestion products of blank filters and filters with snow particles (**Table R1**).

- *Lines 303-317: The use of "low mobile" and at the same time "volatile" to characterize the behavior of Pb is confusing. If Pb is considered as volatile should it not be highly mobile in comparison to other refractory elements? As the term volatile Pb is used in the entire manuscript this specific behavior should be explained in detail before the discussions.* We agree. In fact, Pb is volatile in the atmosphere (especially during fuel burning) however it is low-mobile in dissolved fraction of rivers and lakes where it is present as large-size ferric colloids (Pokrovsky et al., 2016b). These colloids are much less mobile than the soluble low molecular weight fraction of organic complexes of other divalent metals (Zn, Ni, Cd). We revised the text accordingly (L 306-308).

- *Lines 369-370: not agree for Cd, Pb, Sb, Cu and As. Looking at figure, they present a high impact of snow melt water on river for the three latitude zones.* We thank the reviewer for pointing this out and we revised the text accordingly (L 374-377): The impact of snow melt on river export fluxes in spring strongly increases northward for DIC, Cl-, $SO_4^{2-}$, Na, Mg, Ca, Cr, Ni, Mo, Rb, U whereas Cd, Pb, Sb, Cu, As, W and Cs present a high impact of snow melt water on river for the three latitude zones (Fig. 7)."

- *Lines 394-398: Is it not due to the fact that for these elements, the total concentration in the particulate fraction cannot be correlated to their concentration in dissolved fraction, but should be related to specific labile pools that constitute the mineral fraction? Those latter pools being mainly observed using selective extractions. Especially if most of trace elements are supplied by clay mineral (line 430)?* This is a pertinent remark. We added this alternative explanation in the revised text (L 404 - 406). Unfortunately, we could not run selective extractions on very small amount of solid particles in the WSL snow available in this study.

We thank the reviewer for his/her constructive comments. Care of these and other self-motivated corrections we hope the manuscript can meet the high standards of the journal.

Hope to hear from you soon,
Yours Sincerely,                    Oleg S. Pokrovsky

**Table R1**. Compilation of average major and TE concentration in digestion products of blank filters and filters with snow particles.

| Element | Blank filter, µg/L Average (N = 6) | Snow Particles, µg/L Average ( N = 40) |
|---|---|---|
| Li | 0.0012 | 0.091 |
| Be | 0.0000073 | 0.00055 |
| B | 0.78 | 57 |
| Na | 11.6 | 852 |
| Mg | 0.098 | 7.12 |
| Al | 0.50 | 36 |
| Si | 12 | 859 |
| P | 0.086 | 6.29 |
| K | 0.70 | 51.2 |
| Ca | 1.2 | 89.7 |
| Ti | 0.014 | 1.02 |
| V | 0.0038 | 0.279 |
| Cr | 0.058 | 4.25 |
| Mn | 0.015 | 1.09 |
| Fe | 0.216 | 15.8 |
| Co | 0.0004 | 0.0307 |
| Ni | 0.0143 | 1.05 |
| Cu | 0.0063 | 0.459 |
| Zn | 0.0686 | 5.026 |
| Ga | 0.00013 | 0.0094 |
| Ge | 0.00006 | 0.0043 |
| As | 0.00044 | 0.0324 |
| Rb | 0.00051 | 0.0373 |
| Sr | 0.0093 | 0.681 |
| Y | 0.00011 | 0.0083 |
| Zr | 0.0046 | 0.335 |
| Nb | 0.00019 | 0.0141 |
| Mo | 0.0024 | 0.173 |
| Cd | 0.00012 | 0.0086 |
| Sn | 0.0032 | 0.236 |
| Sb | 0.0051 | 0.382 |
| Te | 0.00006 | 0.0045 |
| Cs | 0.000023 | 0.0017 |
| Ba | 0.020 | 1.48 |
| La | 0.00029 | 0.0212 |
| Ce | 0.00047 | 0.0348 |
| Pr | 0.000034 | 0.0025 |
| Nd | 0.00033 | 0.0242 |
| Sm | 0.000025 | 0.0018 |
| Eu | 0.000008 | 0.0006 |
| Gd | 0.000030 | 0.0022 |
| Dy | 0.000025 | 0.0018 |
| Ho | 0.000004 | 0.0003 |
| Er | 0.000013 | 0.0009 |
| Tm | 0.000001 | 0.00011 |
| Yb | 0.000011 | 0.00081 |
| Lu | 0.000001 | 0.00010 |
| Hf | 0.000080 | 0.0058 |
| Ta | 0.0011 | 0.083 |
| W | 0.00055 | 0.0404 |
| Tl | 0.000004 | 0.0003 |
| Pb | 0.0034 | 0.251 |
| Th | 0.000078 | 0.0057 |
| U | 0.000029 | 0.0021 |

---

## Author Response (AR3)

Dear Editor

We carefully corrected all minor English grammar and style remarks, given in the pdf file.

Thank you very much for your thorough reading and correction of the manuscript.

Oleg Pokrovsky